# VICtoR: Learning Hierarchical Vision-Instruction Correlation Rewards for Long-horizon Manipulation

**Kuo-Han Hung**[*]
National Taiwan University

**Pang-Chi Lo**[*]
National Taiwan University

**Jia-Fong Yeh**[*]
National Taiwan University

**Han-Yuan Hsu**
National Taiwan University

**Yi-Ting Chen**
National Yang Ming
Chiao Tung University

**Winston H. Hsu**
National Taiwan University

## Abstract

We study reward models for long-horizon manipulation by learning from action-free videos and language instructions, which we term the visual-instruction correlation (VIC) problem. Existing VIC methods face challenges in learning rewards for long-horizon tasks due to their lack of sub-stage awareness, difficulty in modeling task complexities, and inadequate object state estimation. To address these challenges, we introduce VICtoR, a novel hierarchical VIC reward model capable of providing effective reward signals for long-horizon manipulation tasks. Trained solely on primitive motion demonstrations, VICtoR effectively provides precise reward signals for long-horizon tasks by assessing task progress at various stages using a novel stage detector and motion progress evaluator. We conducted extensive experiments in both simulated and real-world datasets. The results suggest that VICtoR outperformed the best existing methods, achieving a 43% improvement in success rates for long-horizon tasks. Our project page can be found at `https://cmlab-victor.github.io/cmlab-vicotor.github.io/`.

## 1 Introduction

Reinforcement learning (RL) has been extensively studied for long-horizon manipulation [1; 2; 3]. However, crafting reward functions for these tasks is complex, as it typically requires access to true states or significant domain expertise. Consequently, there is an urgent need for a robust and accurate reward model for these tasks. Previous research has explored modeling reward functions via robotic expert demonstrations [4; 5], goal-images [6; 7], and human demonstrations [8; 7; 9]. Unfortunately, the required task specification materials, such as goal-images, remain costly and impractical.

Recently, methods that consider the vision-instruction correlation (VIC) as reward signals have emerged, providing a more accessible way for task specification through language. Specifically, these VIC methods view the reward modeling as a regression or classification problem and train the reward model on the given **action-free** demos and instructions. Pioneer approaches [10; 11; 12; 13; 14] studied using pre-trained Vision Language Models (VLMs) or Video Language Models, like CLIP [15], to generate rewards by assessing the similarity between visual observations and language goals. Additionally, Ma et al. [16] explore the possibility of using human video and language data to pre-train a reward model, later fine-tuned with in-domain data. However, regardless of whether diverse data is used for pre-training, existing VIC studies still require full-length data to retrain the model. Notably, these methods are constrained to short-horizon tasks, such as "*picking up a block*".

Figure 1 illustrates three challenges we observe when applying existing VIC methods to long-horizon manipulation tasks: (1) **No awareness of task decomposition:** Failing to divide complex tasks into manageable parts limits adaptability. (2) **Confusion from variance in task difficulties:** Training a reward model on long-horizon tasks impairs the learning of reward signals and fails to generate suitable progressive rewards. (3) **Ambiguity from lacking explicit object state estimates:** Relying on whole-scale image observations can overlook critical environmental changes. For instance, when

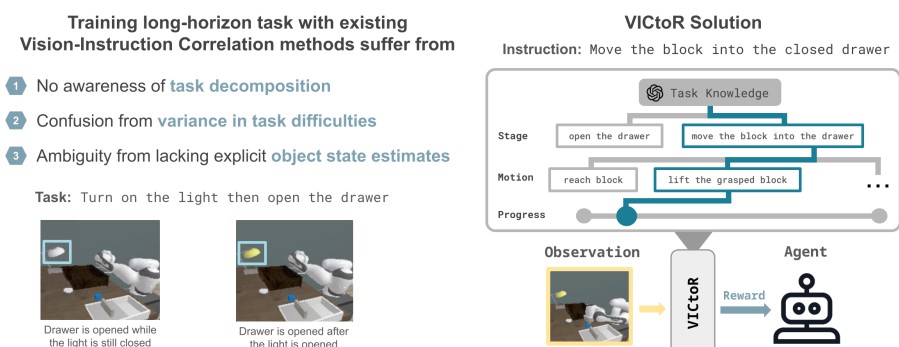

Figure 1: **Problems in existing VIC methods and VICtoR's solution.** Training long-horizon task with existing VIC methods commonly suffer from the listed problems. To address these problems, we propose VICtoR, a hierarchical reward model that can decompose long-horizon tasks and assign rewards by identifying the stage, motion, and progress of the agent from visual observations.

training for the task "*move the block into the closed drawer*", previous VIC models would assign high rewards for moving the block even if the drawer is closed, misleading the learning process.

We propose VICtoR, a hierarchical VIC reward model for long-horizon manipulation. Figures 1 and 2 depict its concept and architecture. VICtoR learns effective rewards for long-horizon tasks by hierarchically assessing overall progress. Specifically, it decomposes long-horizon tasks (high-level) into **stages** (mid-level) and **motions and progress** (low-level, robot primitives), determining the reward by considering task progress at different granularities. Meanwhile, the hierarchical feature sets VICtoR free from the requirements of full-length demonstrations and further grants better generalizability. By treating each long-horizon task as an arbitrary permutation of available motions, VICtoR, trained only on **action-free** motion videos, can be adapted to unseen long-horizon tasks.

To minimize human effort, we leverage GPT-4's semantic understanding to extract task knowledge (expected object statuses and required motions). Then, a stage detector retrieves the current object statuses from the observation and verifies if they match the condition of a specific stage in the task knowledge (Challenge #1). Next, from a list of required motions to complete the stage, a motion progress evaluator determines the current motion and assesses the in-motion progress (Challenges #2 and #3) by evaluating the correlations between visual observations and each motion instruction. Consequently, VICtoR can generate informative rewards to complete long-horizon tasks.

We conducted extensive experiments in both simulated and real-world settings. In the simulation, we designed ten tasks of varying horizons, including those where each step's success depends on the success of the previous stage. For the real-world experiments, we used the dataset collected from XSkill [17] and visualized the trained reward curves in comparison to other baselines. Compared to the best prior VIC reward model, VICtoR achieved a 25% improvement in the average success rate across all tasks and a 43% improvement in the more challenging tasks (tasks which consisted of more than 4 motions). Additionally, our detailed ablation studies emphasized the importance of our hierarchical architecture and training objectives. Visualizations of the learned embedding space and rewards demonstrated that VICtoR effectively assesses task progress.

We summarize our contributions as follows: (1) We are the first to explore the potential of VIC rewards for long-horizon manipulation tasks. (2) We introduce VICtoR, a novel hierarchical VIC reward model that assesses task progress by decomposing it into various levels. (3) We present extensive results from simulated and real-world experiments, ablations, and visualizations to demonstrate VICtoR's superiority, outperforming the best prior method by 43% on more challenging tasks.

## 2 RELATED WORK

**Large Pre-trained Models for Robotics.** With great ability of reasoning, Large Language Models (LLMs) have been utilized for robotics in navigation [18], task planning [19; 20], code-completion

for policies [21; 22], and manipulation [23; 24; 25; 26; 27]. These works leverage LLMs' ability to translate high-level instructions into actionable sequences and break them down into sub-orders for precise agent control. Additionally, recent advancements have leveraged Vision-Language Models (VLMs) to enhance environmental understanding for agents. With a profound understanding of the physical world, VLMs facilitate the works in navigation, robotic control, and monitoring [28; 29; 30; 31; 32]. Unlike previous methods, for reward modeling, VICtoR uses LLMs to break down tasks and VLMs to assess visual observation with motion descriptions, allowing our method to better assess the task's progress and distinguish it with different granularity.

**Addressing Long-Horizon Tasks.** Long-horizon manipulation is a long-standing problem. Traditional Hierarchical Reinforcement Learning (HRL) methods [33; 34; 2; 35; 36] address this challenge by decomposing complex tasks into hierarchical levels or sequential orders, where higher-level policies make decisions that guide the selection of parameters for lower-level policies or pre-trained actions. Task and Motion Planning (TAMP) methods [37; 38; 39] attempt to assemble and schedule well-trained motions to accomplish long-horizon tasks. Additionally, Long-horizon Imitation Learning [40; 41] has been proposed to learn long-horizon tasks from expert demonstrations. Unlike these approaches, VICtoR aims to learn the reward model for long-horizon tasks from action-free demonstrations and task instructions. Our reward model is designed to be independent of policy design and can be applied to RL algorithms to learn long-horizon manipulation tasks.

## 3 PRELIMINARIES

**Problem Statements.** For a reinforcement learning task $\mu$, we assume the environment follows a partially observable Markov Decision Process (POMDP), which can be described with the tuple $\mathcal{M}_\mu := (\mathcal{S}, \mathcal{A}, \mathcal{O}, \Omega, \mathcal{P}, \mathcal{R}, \gamma)$, in which the observation $o \in \Omega$ is derived from the observation function $\mathcal{O}(o \mid s)$ conditioned on the current state $s \in \mathcal{S}$. $\mathcal{A}$ is the space of actions; $\mathcal{P}(s' \mid s, a)$ denotes the dynamics of the environment; $\mathcal{R}$ and $\gamma$ are the environment's reward function and discount factor, respectively. The objective aims to seek a policy $\pi(a \mid o)$ that assigns a probability to action $a$, given the observation $o$, which maximizes the accumulated discounted rewards in a rollout.

**VIC Reward Model.** This work aims to develop a Vision-Instruction Correlation (VIC) reward model to generate accurate rewards from visual observations and language instructions. We assume an instruction $L^\mu$ for the task $\mu$ is accessible, as the instruction is a low-cost resource to collect. Specifically, we seek a reward model $\mathcal{R}(o_t, o_{t-1}, L^\mu)$ computes the reward by evaluating the difference in correlation between the image observation $o$ and the instruction $L^\mu$. Intuitively, $\mathcal{R}(o_t, o_{t-1}, L^\mu)$ should generate a higher reward if the $o_t$ is moving closer to the task's goal compared to $o_{t-1}$.

**Stages and Motions.** In the context of long-horizon manipulation, we define **motion** as the primitive movement that a robot can make, such as reaching or grasping an object. In addition, **stage** refers to a high-level interaction with an object in the environment, requiring the consideration of complex objects' status, and can encompass multiple motions. For instance, the stage "open the drawer" involves the motions "reach the drawer handle" and "pull out the drawer." A comprehensive list of motions and stages is detailed in Appendix C.1.

## 4 METHOD

**Overview.** VICtoR is a reward model for long-horizon manipulation, relying on visual observations and language instructions. As depicted in Figure 2, VICtoR employs a hierarchical approach to assess task progress at various levels, including stage, motion, and motion progress. It consists of three main components: (1) a Task Knowledge Generator that decomposes the task into stages and identifies the necessary object states and motions for each stage (Section 4.1); (2) a Stage Detector that detects object states to determine the current stage based on the generated knowledge (Section 4.2); (3) a Motion Progress Evaluator that assesses motion completion within stages (Section 4.3). With this information, VICtoR then transforms it into rewards (Section 4.4). Both the Stage Detector and Motion Progress Evaluator are trained on motion-level videos labeled with object states, which are autonomously annotated during video collection. This setup enables VICtoR to deliver precise reward signals for complex, unseen long-horizon tasks composed of these motions in any sequence.

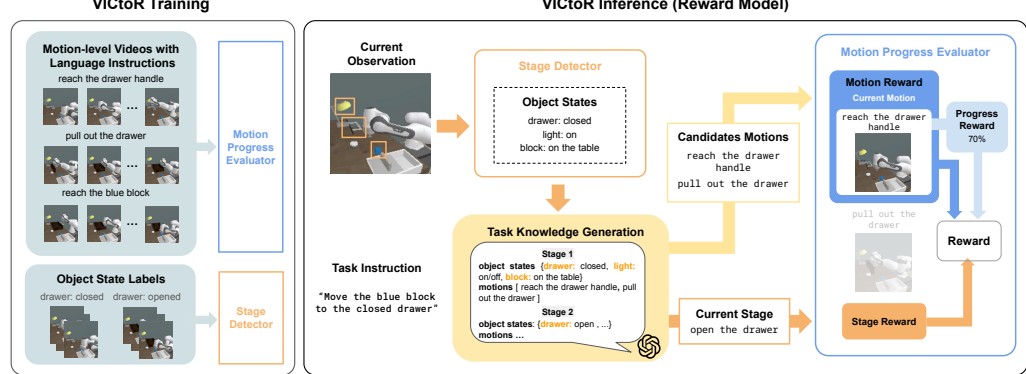

Figure 2: **Training and inference pipeline of VICtoR.** VICtoR is trained using motion-level videos with language annotations and object state labels. It first decomposes the task into task knowledge for decomposed stages, conditional object states, and motions. Then, it uses Stage Detector to identify the stage, and a Motion Progress Evaluator (VLM) to detect the motion and in-motion progress.

## 4.1 TASK KNOWLEDGE GENERATION

To make long-horizon tasks more achievable and manageable, we leverage the reasoning ability in Large Language Models (LLMs) to decompose and analyze the tasks. Previous research [19; 20] introduced LLMs into policy learning by planning sub-goals or actions for explicit guidance to enhance long-horizon capabilities. Building upon this foundation, we explore more nuanced usage. Our goal is to decompose a given training task into distinct sub-stages, define the specific conditions or states of each object at these stages, and identify the motions required by the agent to advance to the next stage. To achieve this, we employ GPT-4 to analyze and construct task knowledge. Figure 2 illustrates an example of task knowledge generation for the task "*move the block into the closed drawer*." Details of our prompt design and implementation are provided in Appendix G, and additional experiments for robustness are described in Appendix B.2.

## 4.2 STAGE DETECTION

Traditional VIC models, which encode entire visual observations to assess task progress, often overlook subtle yet critical changes in object states. To address this, we differentiate between detecting local object state changes and global agent movements (discussed in the next section). In the previous section, Task Knowledge Generator decomposed long-horizon tasks into sub-stages, each conditioned on different object states. To detect these stages, we designed Stage Detector, which detects stage by identifying each object's state in the environment and comparing these with the conditioned states of each stage as defined in the task knowledge.

**Object State Detection.** To detect object states, our Stage Detector first leverages a language-conditioned object detector $D$, such as MDETR [42], to identify objects in the image observation $o$. Essentially, we crop out the bounding box $D(o, c)$ generated from the detector $D$ based on the current observation $o$ and the queried object $c$. Subsequently, these cropped images are fed into our object state classifier $P$. The classifier predicts the likelihood of categorical object state for each cropped image as $P(L^c|D(o, c))$, where $L^c$ represents the language description of possible object states for object $c$. We summarize the training details in Appendix C.2.

**Stage Detection.** After extracting all the object states in the environment, our Stage Detector can then determine the current stage by comparing these states with the task knowledge. A stage is identified if, and only if, all object states align with the task knowledge for a specific stage. For example, as illustrated in Figure 2, if the drawer is currently closed, we compare this condition with the states specified in each stage of the task knowledge. Upon finding that the drawer is expected to be closed only at the first stage, we conclude that the agent is at the first stage. This comparison

should be applied to every conditioned object for the tasks. If the extracted set of object states does not match any stage, the agent will be considered to be in the initial stage.

## 4.3 MOTION PROGRESS EVALUATOR

Since a stage may require multiple motions to complete, relying merely on the stage reward is insufficient for learning complex tasks. To this end, we introduce the Motion Progress Evaluator (MPE) to provide the reward signals within the stage. As discussed in Section 4.1, each detected stage comes with a list of motion descriptions in the task knowledge. Building on this, the MPE assesses the agent's current motion and progress within the motion, taking into account these descriptions. This MPE model yields a more detailed reward signal that accurately reflects progress within stages.

**Architecture.** Our MPE model employs the vision encoder $EV_\theta$ and language encoder $EL_\theta$ from CLIP[15], with two additional heads (fully-connected layers) attached to $EV_\theta$. Initially, the vision encoder $EV_\theta$ generates the base embedding $f$, which is then transformed into motion-specific embedding $m$ and progress-specific embedding $p$ through separate heads. As for the language encoder $EL_\theta$, we employ it to generate the language embedding $l^{mo}$ of motion $mo$ without updating its weights during training. Our goal is to determine the motion class and motion progress by evaluating the similarity between the embeddings $m$ and $p$ and the language description $l^{mo}$, respectively.

**Objectives.** To effectively evaluate progress within a stage, we expect three key capabilities in the MPE model: (1) the ability to discern the temporal difference of frames extracted from the same video, (2) the capacity to identify the motion in which the agent is engaged, and (3) the capability to assess the progress of motion completion. To this end, we introduce objectives based on InfoNCE [43] to guide our MPE model, focusing on time contrast, motion contrast, and language-frame contrast. The functions of the three objectives are illustrated in Figure 7 in Appendix.

For implementation, we utilize the negative L2 distance as the similarity function $S$. During each training iteration, we sample $B$ videos focusing on specific motions and $N$ videos featuring arbitrary agent movements, from which we randomly select a batch of frame sequences $[F_i, F_{j>i}, F_{k>j}]$.

**Time Contrastive Loss.** To capture features relevant to physical interaction and sequential decision-making, we apply the time contrastive loss $\mathcal{L}_{tcn}$ to the base embeddings, as inspired by [44; 45]. Essentially, images that are temporally closer should have more similar representations (embeddings) than those that are temporally distant or from different videos. The loss $\mathcal{L}_{tcn}$ can be formulated by

$$\mathcal{L}_{tcn} = -\sum_{b \in B} \sum_{\substack{(x,y) \in \\ \{(i,j),(j,k)\}}} \log \frac{e^{S(f_x^b, f_y^b)}}{e^{S(f_x^b, f_y^b)} + e^{S(f_i^b, f_k^b)}} \, , \tag{1}$$

where $f_x^b$ is the base embedding of the $x$'th frame.

**Motion Contrastive Loss.** To differentiate between various motions, we introduce the motion contrastive loss $\mathcal{L}_{mcn}$, which aligns each motion's embedding with its relevant language embedding and separates it from unrelated language embeddings. Additionally, to reduce reward hacking [1] and enhance the robustness of our MPE model, we task it with distinguishing frames from motion videos or meaningless videos that contain arbitrary movements. The loss $\mathcal{L}_{mcn}$ is formulated by

$$\mathcal{L}_{mcn} = -\sum_{b \in \{B,N\}} \sum_{\substack{x \in \\ \{i,j,k\}}} \log \frac{e^{S(m_x^b, l^b)}}{e^{S(m_x^b, l^b)} + e^{S(m_x^b, l^{\neq b})}} \, , \tag{2}$$

where $m_x^b$ is the motion embedding of $x^{th}$ frame, $l^b$ is the language annotation of motion in the video $b$, and $l^{\neq b}$ is the instruction for other motion.

---

[1]During RL training, agents may explore areas not covered in demonstrations and exploit reward model vulnerabilities, leading to high rewards for incorrect actions.

**Language-Frame Contrastive Loss.** To assess progress within the motion, we employ the language-frame contrastive loss $\mathcal{L}_{lfcn}$. The loss $\mathcal{L}_{lfcn}$ aims to bring the progress embeddings of nearly completed steps closer to the instruction embedding of the motion while distancing the progress embeddings of frames from earlier steps, which can be stated by

$$\mathcal{L}_{lfcn} = -\sum_{b \in B} \log \frac{e^{S(p_k^b, l^b)}}{e^{S(p_i^b, l^b)} + e^{S(p_j^b, l^b)} + e^{S(p_k^b, l^b)}} , \tag{3}$$

where $p_x^b$ is the progress embedding of $x^{th}$ frame, and $l^b$ is the motion instruction for the video $b$.

**Total Loss.** Finally, we employ the Adam optimizer and the total loss $L_{total}$ to guide our MPE model, where $L_{total}$ is a weighted combination of the three objectives mentioned above:

$$\mathcal{L}_{total} = \lambda_1 \mathcal{L}_{tcn} + \lambda_2 \mathcal{L}_{mcn} + \lambda_3 \mathcal{L}_{lfcn}. \tag{4}$$

**Inference.** During inference, along with the stage detected by the Stage Detector, the MPE model evaluates the current motion from the motion candidates $L^M$ within the stage, provided by the task knowledge from Section 4.1. Specifically, it calculates the similarity scores between the motion embedding $m$ of current observation and language embeddings $l^{mo}$ of each motion candidate $mo \in L^M$. It then selects the motion with the highest score, as formulated by

$$\text{motion}* = \underset{mo \in L^M}{\arg\max} \, S(m, l^{mo}). \tag{5}$$

To determine the confidence level of the MPE model, we further calculate the confidence score by comparing the similarity difference between the motion embedding $m$ and the language embedding $l^{\text{motion}*}$ of the selected motion, and the language embedding $l^n$ of arbitrary agent movements. We decide whether to choose the selected motion $\text{motion}*$ or the first motion by

$$\text{motion} = \begin{cases} \text{motion}*, & \text{if confidence} > \lambda_c \\ 0, & \text{otherwise} \end{cases}, \text{ where} \tag{6}$$

$$\text{confidence} = \frac{S(m, l^{\text{motion}*})}{S(m, l^{\text{motion}*}) + S(m, l^n)}. \tag{7}$$

Finally, given the determined $\text{motion}$, we assess the progress within the motion by calculating the similarity score between the progress embedding $p$ and the motion's language embedding $l^{\text{motion}}$:

$$\text{progress} = \begin{cases} S(p, l^{\text{motion}}), & \text{if confidence} > \lambda_c \\ 0, & \text{otherwise} \end{cases}, \tag{8}$$

## 4.4 REWARD FORMULATION & POLICY LEARNING

Considering all the information acquired from previous sections, we now have knowledge regarding the current stage of the agent, its ongoing motion within the stage, and the progress within that motion. We aggregate this information and the task knowledge into a measure of the task's potential, representing the overall progress within the task:

$$\phi(o_t) = \underbrace{\lambda_m \left( \sum_{i=0}^{\text{stage}} \#\text{motions}_i + \text{motion} \right)}_{\text{Total number of preceding motions}} + \underbrace{\text{progress}}_{\text{in-motion progress}} \tag{9}$$

where $\phi(o_t)$ denotes a potential function conditioned on the image observation at timestep $t$, stage is the number of the detected stage in the task. $\#\text{motions}_i$ is an integer indicating the total number of

different motions within stage $i$, motion is the integer representing the number of the detected motion in the stage, and $\lambda_m$ is a constant near the maximum of progress, ensuring $\phi(o_t)$ increases across motions and stages. Additionally, progress is a float determined by the MPE model. Note that the number of stage, $\#\text{motions}_i$, motion correspond to the sequence provided in the task knowledge. Using the potential, we implemented potential-based shaping rewards $R(o_t, o_{t-1}) = \phi(o_t) - \phi(o_{t-1})$ from [46] to guide the agent's behavior based on potential changes.

## 5 EXPERIMENTS

Our experiments aim to answer the following questions: (1) Does VICtoR provide effective rewards for long-horizon tasks? (2) Is VICtoR able to generate dense rewards from real-world videos? (3) Are all reward signals for stage, motion, and progress indispensable for learning effective rewards in long-horizon manipulation tasks, and do they capture accurate information?

### 5.1 EXPERIMENT SETTINGS

**Evaluation Settings.** Most existing manipulation benchmarks [47; 48; 49] do not support long-horizon tasks where the success of each step depends on the outcomes of previous steps. While Calvin [50] addresses the aforementioned issue, the tasks and action spaces (e.g., rotate) are too challenging for VIC reward models trained on action-free videos. Consequently, we developed a new environment to test different VIC reward models for long-horizon manipulation. For policy training in this environment, we use a ground-truth object detector to speed up the process, as we need to run multiple tasks with multiple seeds. For real-world experiments, we train and test reward models using the XSkill [17] dataset. Details on the environment, tasks, and data are in Appendix C.1 and Figure 3.

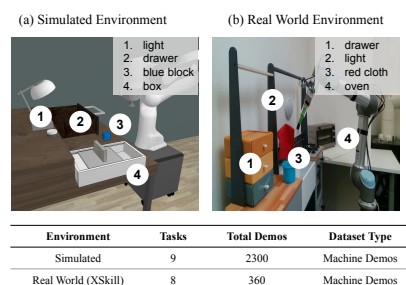

| Environment | Tasks | Total Demos | Dataset Type |
|---|---|---|---|
| Simulated | 9 | 2300 | Machine Demos |
| Real World (XSkill) | 8 | 360 | Machine Demos |

Figure 3: **Environment information.** Tasks are generated from permutations of actions on interactable objects shown in the figure.

**Baselines.** We compare VICtoR with the following baselines: (1) **Sparse Reward:** A binary reward function assigns a reward only when the task succeeds. (2) **Stage Reward:** A reward function that assigns a reward equal to the stage number when the agent reaches a new stage. (3) **LOReL [51]:** A language-conditioned reward model that learns a classifier $f_\theta(o_0, o_t, l)$ to evaluate whether the progression between the $o_0$ and $o_t$ aligns with the task instruction $l$. Note that, we replaced the backbone of LOReL to CLIP for a fair comparison with VICtoR and LIV. (4) **LIV [16]:** A vision-language representation for robotics that can be utilized as a reward model by finetuning on target-domain data. We apply the same reward shaping method for LIV and LOReL as discussed in Section 4.4. (5) **VICtoR (task):** A baseline for VICtoR trains on task-level data, using the same MPE objective but with demonstrations of tasks not divided into stages or motions. Note that the training demos for VICtoR are action-free videos with text instructions; therefore, we do not compare our method with other works requiring action data, such as language-conditioned imitation learning.

### 5.2 POLICY LEARNING WITH VIC REWARDS

In this section, we train the Proximal Policy Optimization algorithm [52] using various reward functions on the simulated environment. We utilize a sparse reward as a signal for task success and shape the reward function using different reward models. Our experiments are divided into two sets: initially, we train on 3 single-stage tasks with varying numbers of motions, and subsequently, on 7 tasks with varying numbers of stages. As shown in Tables 1 and 2, VICtoR consistently outperforms current baselines, achieving a 25% average performance gain. This advantage increases to 43% in harder tasks with more than three motions, highlighting VICtoR's effectiveness in training for long-horizon tasks. The significant benefits of the motion design are also evident when compared to the task-level baseline, especially in more complex tasks. Furthermore, as Table 1 demonstrates, even at the task level, VICtoR outperforms other baselines, showcasing the effectiveness of our training objective detailed in Section 4.3. Training details and curves are provided in Appendix C.4.

Table 1: **Experiment of tasks with one stage but different numbers of motions.** The table presents the success rate [↑] of the learned policy using different reward functions across three tasks that have one stage but vary in the number of motions. It shows that the differences between VICtoR and other baselines become larger as the number of motions in the tasks increases. This indicates that when a stage requires more detailed actions to complete, it becomes necessary to decompose the task to assess progress. Results are averaged over three different seeds.

| #motions | 1 | 2 | 4 |
|---|---|---|---|
| Task | reach block | pick block | move block |
| Sparse | 99.8% | 0.0% | 0.0% |
| LIV [16] | 99.9% | 98.6% | 0.0% |
| LOReL [51] | **100.0%** | 88.8% | 0.0% |
| VICtoR(task) | 99.9% | 94.2% | 36.8% |
| **VICtoR** | 99.9% | **98.8%** | **58.9%** |

Table 2: **Experiment of tasks with different numbers of stages and motions.** Extending from Table 1, this table includes seven additional tasks with varying numbers of stages and motions. It shows that VICtoR outperforms prior methods, and the differences become larger as the complexity of the tasks increases, demonstrating the effectiveness of our design of stage and motion rewards. Results are averaged over three different seeds.

| #stages | 1 | 1 | 2 | 2 | 2 | 3 | 3 |
|---|---|---|---|---|---|---|---|
| #motions | 2 | 2 | 3 | 4 | 6 | 4 | 5 |
| Task | open box | open drawer | open drawer + open light | open box + pick block | open drawer + move block | open light + open drawer + reach block | open box + open light + pick block |
| Sparse | **100.0%** | 79.5% | 0.0% | 0.0% | 0.0% | 0.0% | 0.0% |
| Stage | **100.0%** | 79.5% | 26.0% | 0.0% | 0.0% | 0.0% | 0.0% |
| LIV [16] | 99.9% | 66.7% | 0.0% | 0.0% | 0.0% | 0.0% | 0.0% |
| LOReL [51] | **100.0%** | **100.0%** | 33.1% | 35.1% | 0.0% | 35.0% | 28.9% |
| **VICtoR** | 99.9% | **100.0%** | **66.1%** | **96.4%** | **30.6%** | **70.9%** | **57.7%** |

## 5.3 ABLATION STUDIES

To better understand how each reward signal in VICtoR contributes to policy learning, we conducted an ablation study on the "*open box then pick blue block*" task to evaluate the effectiveness of VICtoR's stage determination, motion determination, and progress assessment as introduced in Section 4. Based on the findings presented in Table 3, we observe that injecting reward signals at each level proves beneficial for policy learning. Furthermore, our analysis indicates that reward signals at the "*progress*" level notably enhance performance. This underscores the importance of nuanced reward signals, particularly in tasks with longer horizons, thereby supporting our claims. Ultimately, the combination of these three levels of reward signals enables the policy to achieve peak performance.

Table 3: **Ablation study.** This table illustrates the effectiveness of each level of reward signals in VICtoR for policy learning. Results are averaged over three different seeds.

| Ablation Settings | | | Success Rate |
|---|---|---|---|
| Stage | Motion | Progress | |
| - | - | - | 0.0% |
| ✓ | - | - | 22.8% |
| ✓ | ✓ | - | 29.7% |
| ✓ | ✓ | ✓ | **96.4%** |

## 5.4 QUALITATIVE ANALYSIS

We plot the potential curves from three different reward models across four tasks with varying complexities (with longer horizons on the right). The results in Figure 4 demonstrate that VICtoR effectively generates reward curves for complex, composed tasks, while other baseline models fail to

provide progressive signals as task horizons grow. Notably, while prior reward models were trained on full-length, long-horizon demonstrations, VICtoR was only trained on motion-level data yet still achieved the most optimal results. We also put the analysis for misordered action in Appendix B.1.

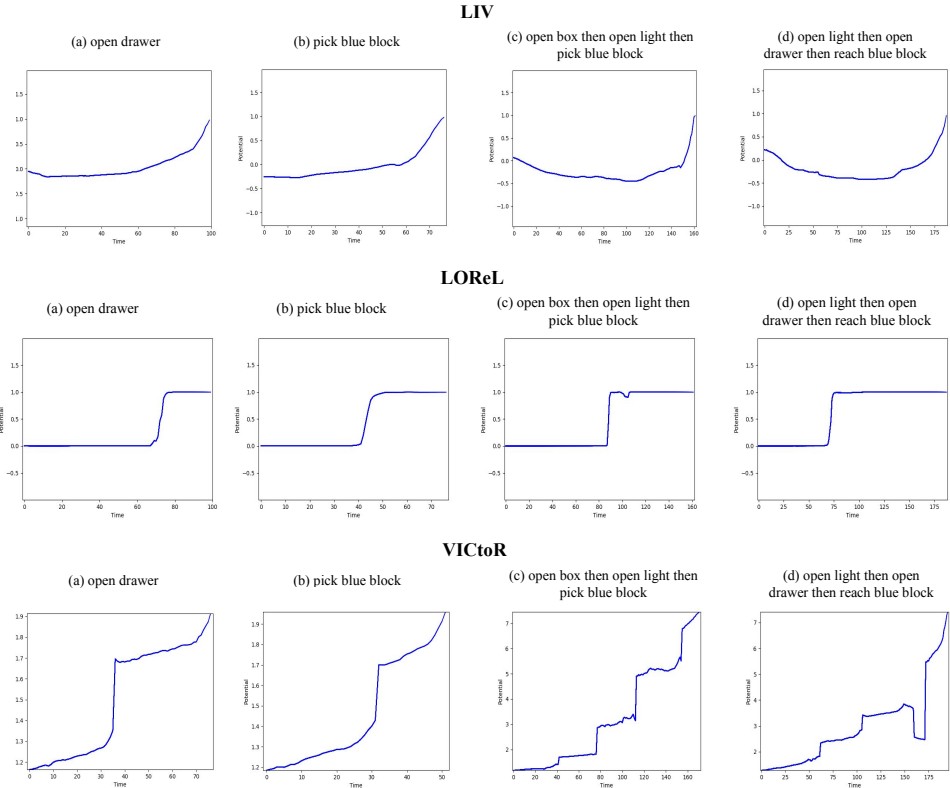

Figure 4: **Potential comparison across different tasks**: We compare the potential generated by different reward models. In these comparisons, we can see that VICtoR provides the most progressive and near-strictly increasing potential function, especially as the horizon increases. This demonstrates its ability to provide fine-grained rewards for long-horizon tasks.

## 5.5 REAL WORLD EXPERIMENTS

To demonstrate VICtoR's practical efficacy, we visualized the potential curves $\phi(o_t)$ for two scenarios: one with videos that match the corresponding instructions, and another with incorrect videos using the same instructions. As illustrated in Figure 5, for the correct actions, the potential curves show that VICtoR effectively identifies task progress in real-world scenarios by increasing the potential as the agent completes tasks, a capability not matched by previous VIC reward models. For the incorrect actions, as the agent moves from the right to the left side to close the drawer, VICtoR's visualization shows an initial increase in potential. This increase is logical as these movements approach the light, aligning with the first stage of the instruction, "*open the light.*" However, as the agent continues toward the drawer, VICtoR recognizes the incorrect task and begins to decrease the reward, effectively deterring the agent's movement. This highlights VICtoR's ability to analyze agent movement and task progress accurately. Additional visualizations for various tasks are provided in Appendix F.

## 5.6 VISUALIZATIONS

**Embedding Visualization.**     To verify the effectiveness of the contrastive objectives we designed to boost understanding of task progress, we present a t-SNE analysis on motion embeddings $m$ across every motion in our simulated environment. From Figure 6, we observe that the learned embeddings

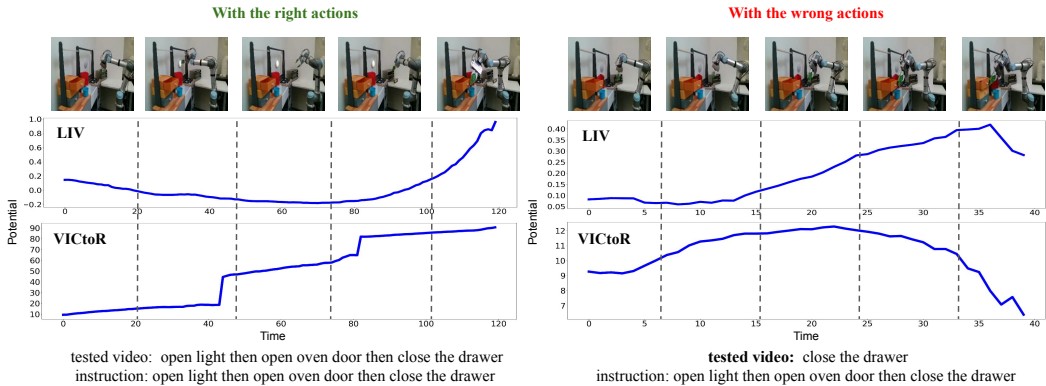

Figure 5: **VICtoR on real world data.** This figure displays LIV [16] and VICtoR's potential visualizations for long-horizon tasks from XSkill with correct and incorrect test videos.

of identical motion are clustered, indicating high discriminative capability. This attribute is crucial for enhancing the accuracy and efficiency of our reward model, particularly in long-horizon tasks.

**Reward Visualization.** To assess VICtoR's ability to differentiate motions and assess progress within a motion, we visualize motion determination per timestep and measure the embedding $z$'s distance between motion descriptions and frame embeddings. As shown in Figure 6, VICtoR accurately switches motions at the appropriate timestep and decreases the embedding distance of the determined motion as the agent approaches each motion's goal.

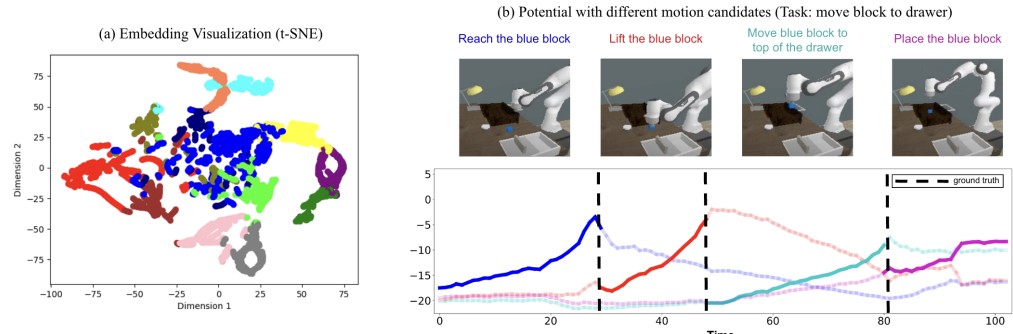

Figure 6: **Visualization.** (a) t-SNE analysis for different motion frames, and (b) analysis of negative embedding distances and determined motions in a demonstration video.

## 6  CONCLUSION

We have presented VICtoR, a Hierarchical Vision-Instruction Correlation Reward Model for Long-horizon Robotic Manipulation. Using only language instructions, VICtoR is able to provide effective rewards to the reinforcement learning agent. Having been trained solely on modular short-horizon demonstration videos, VICtoR outperforms state-of-the-art approaches trained on targeted long-horizon tasks and successfully operates in both simulated and real-world environments.

**Future works and limitations** VICtoR can provide rewards for unseen long-horizon tasks composed of seen motions. However, similar to previous works, the limitation of VICtoR is that it cannot be applied to tasks involving unseen motions. Therefore, a future direction for this project will be training with a diverse range of motion data to explore its zero-shot capabilities on unseen motions.

ACKNOWLEDGEMENTS

This work was supported in part by the National Science and Technology Council, Taiwan, under Grant NSTC 113-2634-F-002-007 and NSTC 113-2813-C-002-031-E. We also thank all reviewers and area chairs for their valuable comments and positive recognition of our work during the review process.

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

# APPENDIX

## A    ILLUSTRATIONS FOR LOSS OBJECTIVES

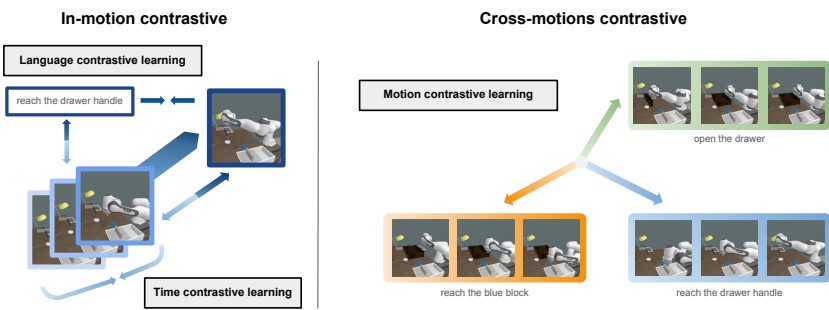

Figure 7: **In-motion and cross-motion Contrastive Objectives:** We illustrate how the three contrastive loss functions we applied manipulate the distances and relationships between instruction representations and encoded visual observations within the embedding space.

## B    ADDITIONAL EXPERIMENTS

### B.1    QUALITATIVE ANALYSIS - MISORDERED ACTION

We visualize the reward by giving text goals for long-horizon tasks (open light then open drawer then reach blue block) on misordered action (reach blue block) videos to see the difference in reward models. The result in Figure 8 indicates that VICtoR is capable of determining the current state of environmental objects to decide the level of potential while LIV simply generates progressive signals on each task.

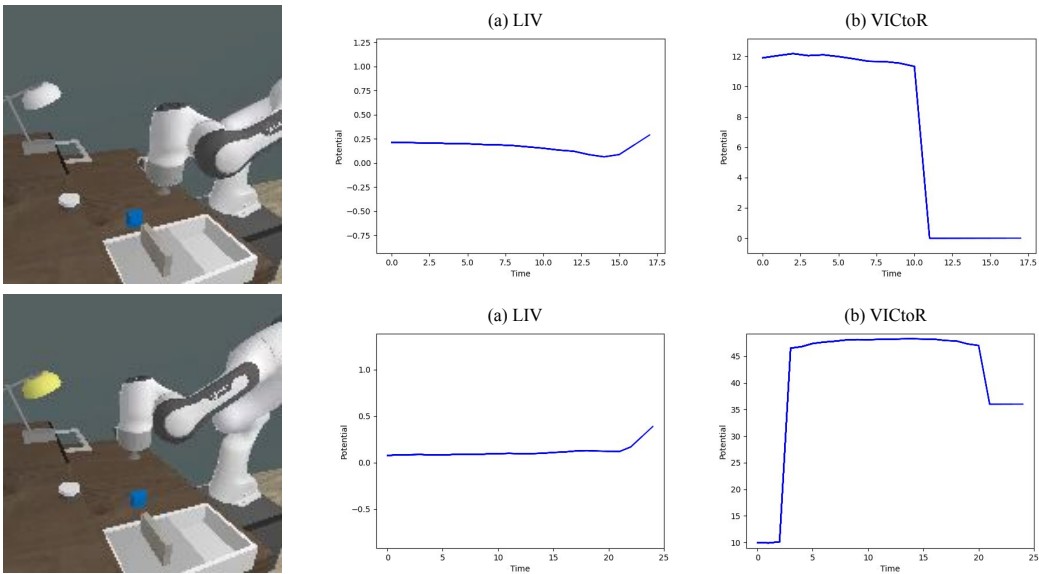

Figure 8: **Potential Visualization on Misordered Action**: We visualize the rewards given for long-horizon tasks (open light, then open drawer, then reach blue block) in videos with misordered actions (reach blue block) to compare different reward models. The figure shows that VICtoR can determine the current state of environmental objects to decide the level of potential, while LIV simply generates progressive signals for each task without penalizing incorrect actions, which may lead to poor learning outcomes.

## B.2 ACCURACY OF TASK KNOWLEDGE GENERATION

To validate the robustness of LLMs for task knowledge generation, we conducted an additional experiment testing the accuracy of LLM outputs. We tested our model by randomly selecting tasks, which could involve 1 to 4 stages. We then used GPT-4 to translate these tasks into human-like queries. With these queries, we tested the task knowledge generation model by comparing its output against the ground truth. After randomly sampling 50 different tasks with varying free-form instructions, we observed that GPT-4 achieved an accuracy of 96% in decomposing tasks into stages, motions, and conditional object states, confirming its robustness for this application.

## B.3 ABLATION STUDY ON THREE CONTRASTIVE LOSSES

To examine the importance and effectiveness of the three proposed contrastive objectives in Eq. 4, we train the Motion Progress Evaluator under three conditionsIn each experiment, we remove one contrastive loss, as illustrated in Table 4. This study is conducted on the "*open box then pick blue block*" task. The results clearly demonstrate that each objective plays a significant role, particularly the $L_{lfcn}$ objective. This objective effectively differentiates the progress in motion, providing nuanced guidance for long-horizon tasks.

Table 4: Ablation Study on the proposed three contrastive losses

| Ablation Settings | | | Success Rate |
| --- | --- | --- | --- |
| $\mathcal{L}_{tcn}$ | $\mathcal{L}_{mcn}$ | $\mathcal{L}_{lfcn}$ | |
| ✓ | ✓ | - | 0.0% |
| ✓ | - | ✓ | 61.8% |
| - | ✓ | ✓ | 94.0% |
| ✓ | ✓ | ✓ | **96.4%** |

## B.4 ABLATION STUDY ON AUTOMATIC ANNOTATION BY LVLMS

Since object status labels are required to deploy VICtoR in a new environment, we conduct an ablation study to automatically annotate these labels using modern LVLMs. The study is performed on the XSkill dataset, focusing on four interactable objects: cloth, light, oven, and drawer. Specifically, for each video where the robot interacts with one of these objects, we first use MDETR (without fine-tuning) to obtain cropped images of the target object. These cropped images are then provided to GPT-4V, which annotates the object's status. Finally, we compare the consistency between human-annotated and LVLM-annotated labels, as summarized in Figure 9.

The results demonstrate a high level of consistency between the two sets of labels, indicating that deploying VICtoR in new environments is highly feasible. Most inconsistent labels arise during fuzzy phases when switching motions. These inconsistencies are easily corrected or even negligible.

## C DETAILED SETTINGS FOR THE EXPERIMENT

### C.1 DETAILS ABOUT OUR SIMULATED ENVIRONMENT AND TASKS

**Details about the environment**  Our environment was developed using Coppeliasim [53], with Pyrep [54] serving as the coding interface, as shown in Figure 3. In this environment, there are four interactable objects: a light, a drawer, a blue block, and a box, each associated with different stages as detailed in Section C.1. The tested tasks can be any permutation of these stages.

**List of Stages and Motions.**  We treated the motions as the action primitives for agents to implement. Definitions of the motions in each stage in our experiments are detailed as shown in Table 5. The definition of motions of stages could be initiated by either humans or GPT.

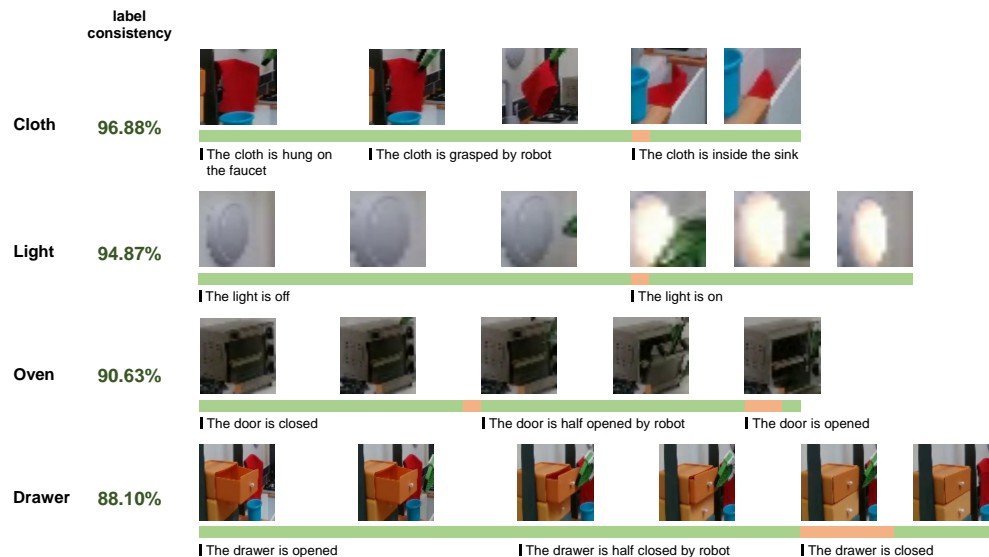

Figure 9: **Automatic Annotation by LVLMs**: We leverage LVLMs (i.e., GPT-4V) to automatically annotate the status of objects in the XSkill dataset. The results are then compared with human-annotated labels to evaluate label consistency. The findings indicate that modern LVLMs already possess the capability to handle this task effectively. Most inconsistencies occur during the fuzzy phases when switching motions.

Table 5: Description of Separated Motions for Each Stage

| Stage | Motion 1 | Motion 2 | Motion 3 | Motion 4 |
|---|---|---|---|---|
| reach blue block | reach the blue block | - | - | - |
| pick blue block | reach the blue block | lift the grasped blue block | - | - |
| move blue block to drawer | reach the blue block | lift the grasped blue block | move the blue block to the top of the drawer | place the blue block down to the drawer |
| open drawer | reach the closed drawer handle top | pull the drawer out | - | - |
| close drawer | reach the opened drawer handle top | push the drawer forward | - | - |
| open box | reach the box holder back | slide the box holder forward | - | - |
| close box | reach the box holder front | slide the box holder backward | - | - |
| open light | reach and push down the button | - | - | - |
| close light | reach and push down the button | - | - | - |

**Training Tasks Details.** The detailed success conditions, training timesteps, and the running machines for each task are shown in Table 6.

**Training Data Details.** To train the reward models for VICtoR and other baselines, we collected a dataset in our environment. Note that VICtoR requires only motion-level data for training, while other baselines need full-length videos. Therefore, the data used to train VICtoR and other baselines differ in type but are comparable in total amount per motion or stage. For VICtoR, we provide only the motion-level demos, with each motion accompanied by 150 demos and corresponding motion

Table 6: Training Details for each task

| | |
|---|---|
| **Reach blue block** | |
| **Task Goal** | Reach the blue block |
| **Success Condition** | The distance between the end-effector and the blue block |
| **Max Training Step** | 30000 |
| **GPU** | NVIDIA RTX 4090 |
| **Pick blue block** | |
| **Task Goal** | Grasp the blue block and lift it up |
| **Success Condition** | The blue block is lifted higher than 0.62 |
| **Max Training Step** | 100000 |
| **GPU** | NVIDIA RTX 4090 |
| **Move blue block to drawer** | |
| **Task Goal** | Pick up the blue block and place it into the drawer |
| **Success Condition** | The position of the blue block (if it is in drawer or not) |
| **Max Training Step** | 300000 |
| **GPU** | NVIDIA RTX 4090 |
| **Open drawer** | |
| **Task Goal** | Pull out the drawer |
| **Success Condition** | The position of the drawer |
| **Max Training Step** | 50000 |
| **GPU** | NVIDIA RTX 4090 |
| **Open box** | |
| **Task Goal** | Push the lid of the box to open the box |
| **Success Condition** | The position of the box lid |
| **Max Training Step** | 50000 |
| **GPU** | NVIDIA RTX 4090 |
| **Open drawer then open light** | |
| **Task Goal** | Pull out the drawer and then press the button to open the light |
| **Success Condition** | The position of the drawer and the color of the light |
| **Max Training Step** | 200000 |
| **GPU** | NVIDIA RTX 3090 |
| **Open box then pick blue block** | |
| **Task Goal** | Push the lid of the box to open it and pick up the blue block |
| **Success Condition** | The position of the box lid and the height of the blue block |
| **Max Training Step** | 200000 |
| **GPU** | NVIDIA RTX 3090 |
| **Open drawer then move blue block to drawer** | |
| **Task Goal** | Pull out the drawer and then pick up the blue block and place it into the drawer |
| **Success Condition** | The position of the blue block (if it is in drawer or not) |
| **Max Training Step** | 400000 |
| **GPU** | NVIDIA RTX 3090 |
| **Open light then open drawer then reach blue block** | |
| **Task Goal** | Press the button to open the light, then pull out the drawer, and reach the blue block |
| **Success Condition** | The color of the light, the position of the drawer, and the distance between the end-effector and the blue block |
| **Max Training Step** | 400000 |
| **GPU** | NVIDIA RTX 3090 |
| **Open box then open light then pick the blue block** | |
| **Task Goal** | Push the lid of the box to open it and press the button to turn on the light then pick up the blue block |
| **Success Condition** | The position of the box lid, the color of the light, and the height of the blue block |
| **Max Training Step** | 400000 |
| **GPU** | NVIDIA RTX 3090 |
| **Open box then open light then open drawer then then reach the blue block** | |
| **Task Goal** | Push the lid of the box to open it and press the button to turn on the light and open the drawer then reach the blue block |
| **Success Condition** | The position of the box lid, the color of the light, the position of the drawer, and the distance between the end-effector and the blue block |
| **Max Training Step** | 500000 |
| **GPU** | NVIDIA RTX 3090 |

descriptions. For training other baseline models, such as LIV [16] and LOReL [51], we provide the **full-length demo videos**, each paired with task descriptions and consisting of 150 demos per task.

## C.2   DETAILS OF STAGE DETECTOR AND TRAINING DETAILS OF MOTION PROGRESS EVALUATOR

The structure of classifier $P$ for Stage Detector includes a CLIP Text Encoder for processing object state descriptions and a CLIP Image Encoder for processing cropped images. These encoders are followed by a downstream MLP classifier that scores relevance.

We provide more details of the Stage Detector and Training Details of the Motion Progress Evaluator in Table 7 and Table 8. For the hyperparameter selection of $\lambda_{1,2,3}$, we tried various combinations but found that the model is not sensitive to these hyperparameters. Therefore, we have chosen to set them all to the same values.

Table 7: MDETR configuration in VICtoR

| Object Detector - MDETR [42] | |
|---|---|
| Backbone | Resnet101 |

Table 8: Details of VICtoR Motion Reward Model and Stage Determinator Training

| Motion Progress Evaluator | |
|---|---|
| Language Encoder | CLIPTextModel |
| Image Encoder | CLIPVisionModel |
| CLIP Model | openai/clip-vit-base-patch32 |
| Learning Rate | 0.000001 |
| Batch Size | 32 |
| Training Steps | 20000 |
| Optimizer | Adam [55] |
| Loss Function | Equation 4 |
| Parameters in Equation 4 | $\lambda_1 = 1, \lambda_2 = 1, \lambda_3 = 1$ |

| Stage Determinator | |
|---|---|
| Language Encoder | CLIPTextModel |
| Image Encoder | CLIPVisionModel |
| CLIP Model | openai/clip-vit-base-patch32 |
| Learning Rate | 0.00001 |
| Batch Size | 32 |
| Training Steps | 40000 |
| Oprimizer | Adam |
| Loss Function | Binary Cross Entropy Loss |

## C.3   DETAILS OF MPE'S INFERENCE AND REWARD FORMULATION

We organize our reward function parameters of Section 4.3 in Table 9, For the selection of $\lambda_m$, we tried to find the biggest range of the progress (embedding distance from language to motion), in order to make the proceeding motion's potential higher than the previous one. For the selection of $\lambda_c$, we have tested it with different numbers and found out that when the $\lambda_c$ is in the range of 0.1 to 0.4, the model has similar results, and if it is less than 0.1, it cannot avoid reward hacking. Once it is larger than 0.4, the model cannot provide enough signal for the agent to learn.

Table 9: Reward formulation and MPE model parameters of VICtoR

| $\lambda_c$ | $\lambda_m$ |
|---|---|
| 0.2 | 36 |

## C.4 Training Details of Policy Training

**Training details.** We trained the policy under the **SubprocVecEnv** from **stable-baseline3** [56] with reward normalization by **VecNormalize**. For online interactable environment, we built a simulation environment with **Coppeliasim** [53] and **Pyrep** [54] and wrapped the environment for policy learning under the framework of **OpenAI Gym** [57]. Training time is about 3 hours for 10000 steps. Training details for each task and policy are summarized in Tables 6 and 10, respectively.

Table 10: Details of policy training

| Policy | PPO [52] |
|---|---|
| Learning Rate | 0.003 |
| Policy Model | MlpPolicy |
| Discount factor ($\gamma$) | 0.99 |
| Reward Normalization | True |

## C.5 Computational cost and Resource requirements

The computational cost in our framework is 2 GPU hours for the training of the progress evaluator. For the policy learning, training with our reward model for 20000 steps required approximately 12 GPU hours. Both of the stats above are running on RTX 4090.

Regarding the inference time of the current implementation, the object detector takes approximately 51.5ms, and the reward model accounts for 24ms on average over 1000 steps (tested on a CPU Intel(R) Core(TM) i7-13700 and a GPU RTX 3090). Considering that end effector action latency is typically long in both simulation and real-world scenarios, we believe that a 0.05s inference time would not significantly impact the agent's efficiency, even in a real-world environment.

## C.6 Training Details of Baseline Models

We provide details of other baselines in Table 11.

Table 11: Details of Baseline Reward Models Training

| LIV | |
|---|---|
| CLIP Backbone | Resnet50 |
| Learning Config | pretrain |
| Batch Size | 32 |
| Train Steps | 70000 |
| Learning Rate | 0.00001 |
| Weight Decay | 0.001 |

| LOReL | |
|---|---|
| Language Encoder | CLIPTextModel |
| Image Encoder | CLIPVisionModel |
| CLIP Model | openai/clip-vit-base-patch32 |
| Batch Size | 32 |
| Train Steps | 20000 |
| Learning Rate | 0.00001 |
| Flipped Negative | True |
| alpha | 0.25 |

## C.7 DETAILS OF EXPERIMENTS ON XSKILL

For the visualization experiments on real-world dataset XSkill [17] in Appendix F, we train VICtoR on 120 demonstrations with each breaking down into 3 different motion-level videos while baseline models are trained with the full-length demonstrations. Details about the 4 motion level skills and the test tasks in the visualization experiments are listed in Table 12.

Table 12: Text description for motions and test tasks in XSkill

| Motion Name | Description |
|---|---|
| close the drawer | push the drawer so that it is closed |
| open the oven door | open the oven door |
| open the light | push the light on |
| move the cloth into the sink | grab the cloth and release it into the sink |
| **Test Tasks** | |
| close the drawer then move the cloth into the sink then open the light | |
| close the drawer then move the cloth into the sink then open the oven door | |
| close the drawer then open the light then move the cloth into the sink | |
| close the drawer then open the light then open the oven door | |
| open the light then move the cloth into the sink then open the oven door | |
| open the light then close the drawer then move the cloth into the sink | |
| open the light then open the oven door then close the drawer | |
| open the oven door then open the light then move the cloth into the sink | |

## D LEARNING CURVES OF POLICY LEARNING

We evaluate the policies trained from different reward functions with the success rate in Figure 10, the shadows show the standard deviation.

## E ADDITIONAL COMPARISONS BETWEEN VICtoR AND OTHER SOLUTIONS TO ROBOTIC POLICY LEARNING

**Imitation Learning.** Prior works have attempted to solve the robot manipulation tasks via imitation learning from either robot demonstrations or human videos. However, in our setting, we **do not use action sequences from demonstrations** to train VICtoR. Thus, the line of imitation learning works is not considered as our baseline. Our objective is to learn a reward model solely from vision and language inputs, without relying on action or observation vectors. This distinction sets our approach apart from imitation learning and other similar methods, which typically require action information to clone a behavior. Therefore, our method addresses a unique problem space that is not directly comparable to imitation learning.

**Motion Decomposition** Previous studies [41; 58] have utilized motion decomposition to facilitate learning in long-horizon manipulation tasks. By breaking down these tasks into subgoals, they demonstrated that imitation policy networks could be trained more effectively. While we also use the concept of task decomposition, our method introduces a tailored framework for VIC reward models, as shown in Figure 1. We segment tasks into three levels: stage, motion, and progress, providing a finer-grained approach to handle task complexities. Our experiments show that this structured method significantly improves the effectiveness of VIC reward models in complex scenarios.

**Task and Motion Planning (TAMP).** Prior works investigated through accomplishing long-horizon robot manipulation tasks via arranging trained skills. While TAMP methods require a predefined and well-trained skill set and put more emphasis on planning for the execution sequence of these trained skills, our model aims to provide reward signals directly toward policy learning to guide the agent to learn a long-horizon policy from scratch with reinforcement algorithms. Therefore, we do not consider TAMP methods as baselines for VICtoR.

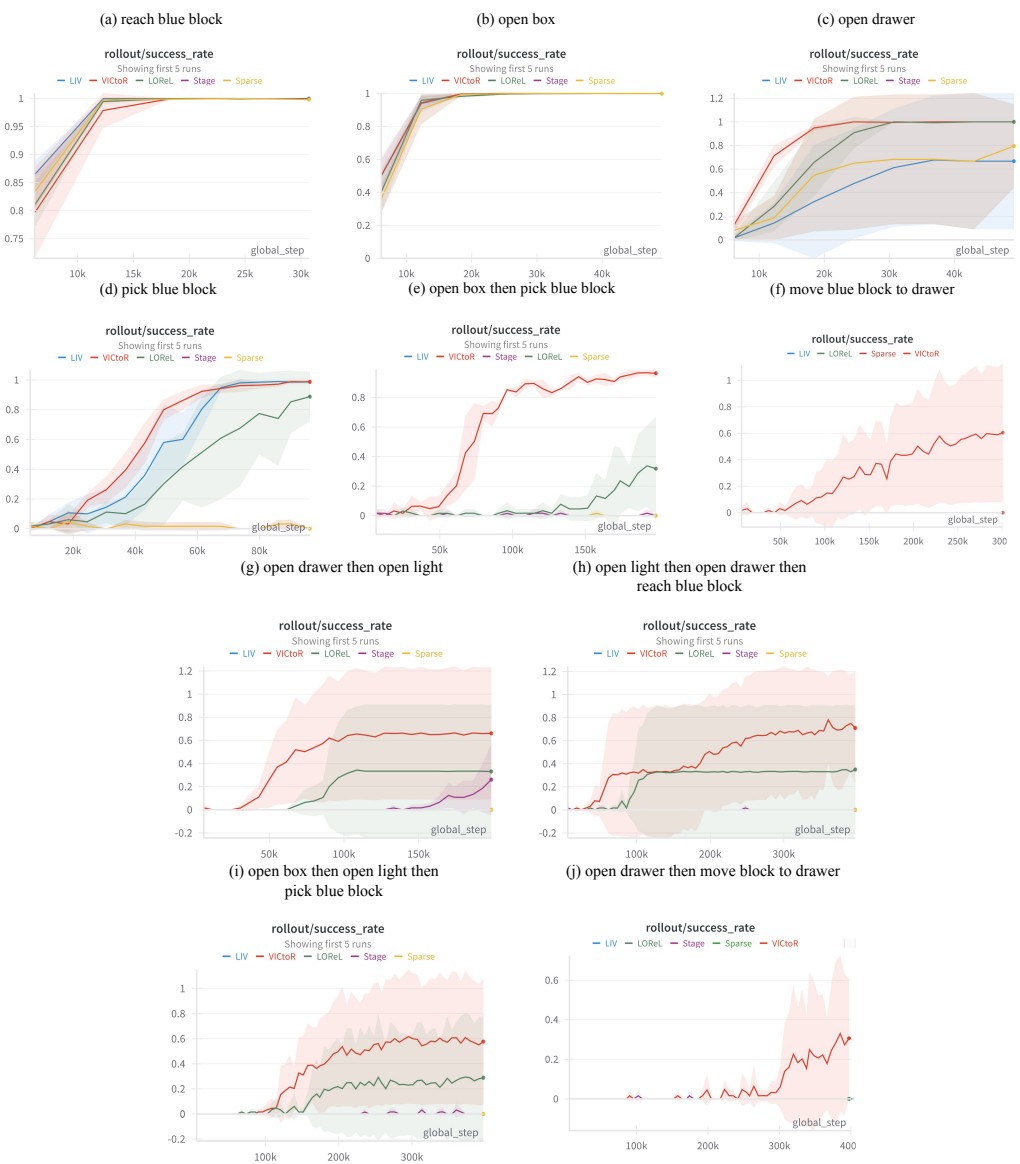

Figure 10: **Training Curve Visualization**: We visualize the success rate across various training episodes and display the success rate curves for different methods in different tasks. The figure demonstrates that VICtoR effectively guides the RL agent to learn quickly and achieve the highest success rate in the fewest episodes, highlighting the effectiveness of the rewards generated by VICtoR.

## F    MORE POTENTIAL VISUALIZATION FOR COMPOSED TASKS IN XSKILL [17]

The visualization results are shown in Figure 11 and Figure 12.

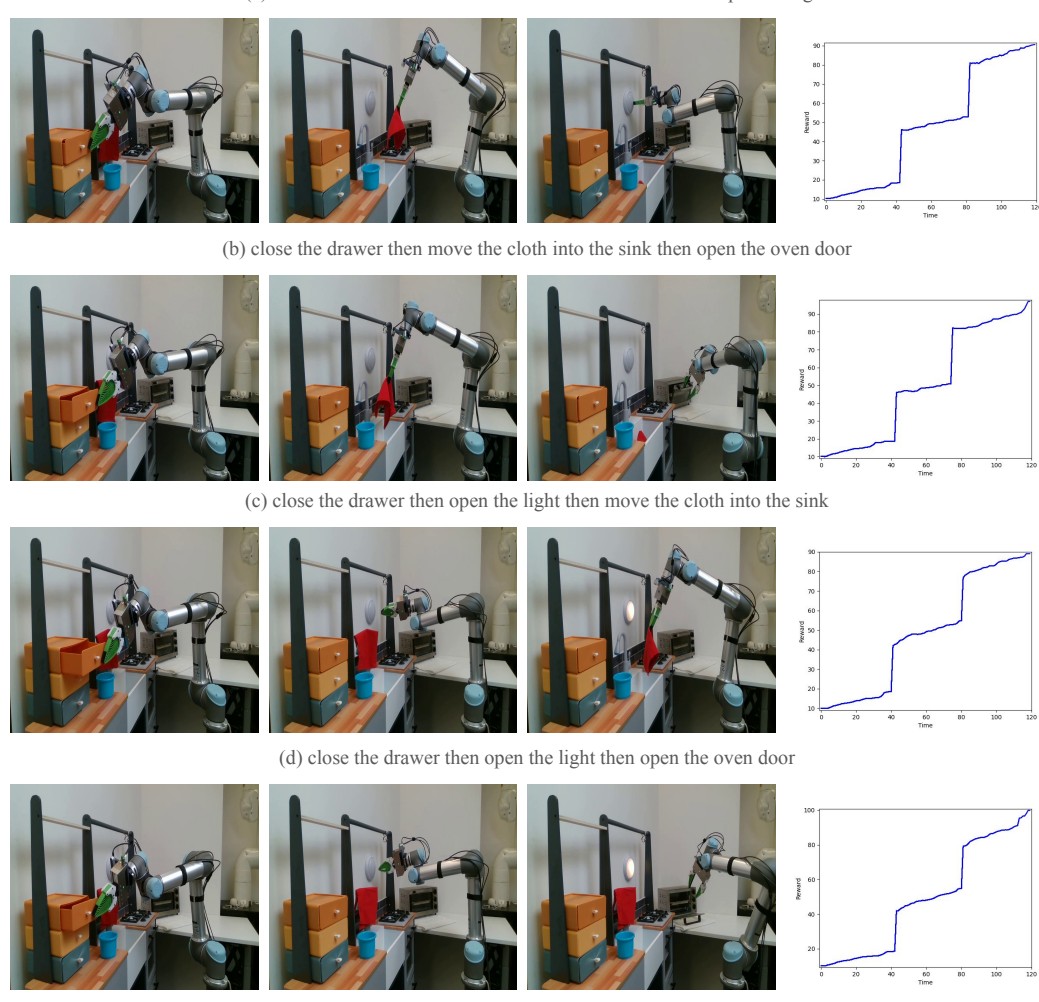

Figure 11: **Potential Visualization on XSkill**: Evaluation of the potential curve across multiple test videos on different tasks for XSkill.

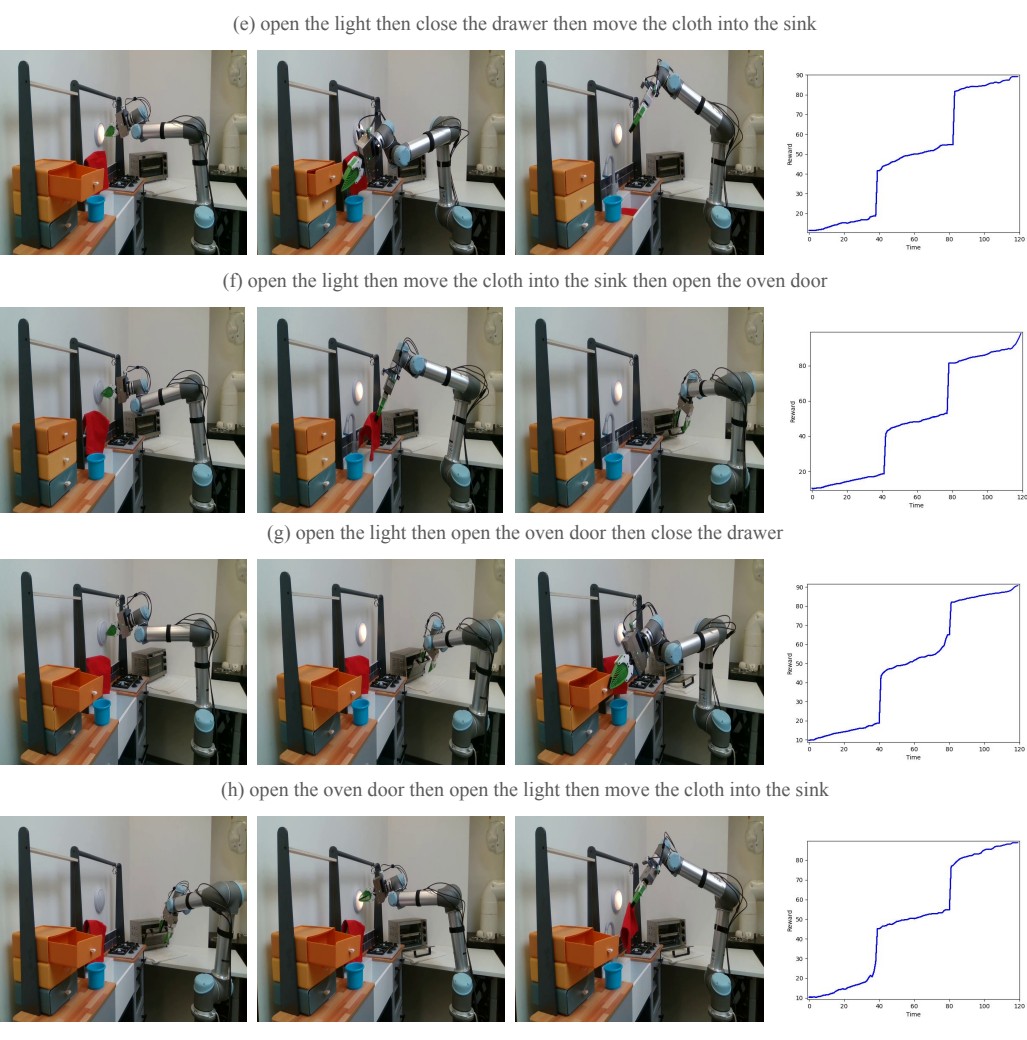

Figure 12: **Potential Visualization on XSkill**: Evaluation of the potential curve across multiple test videos on different tasks for XSkill.

# G PROMPTS FOR TASK KNOWLEDGE GENERATION

```
You are a task splitter. Given a task, you should split it into stages
    ↪ and motions. Also, during transitions between stages, you must
    ↪ provide the environment status.

## Possible Motions

### drawer
- reach the open drawer holder top
- push the drawer forward
- reach the closed drawer holder top
- pull the drawer out

### box (slide forward to open, slide backward to close)
- reach the box holder back
- slide the box holder forward
- reach the box holder front
- slide the box holder backward

### light
- reach and push down the button

### blue_block
- reach the blue block
- lift the grasped blue block
- move the blue block to the top of the drawer
- move the blue block to the top of the table
- place the blue block down to the drawer
- place the blue block down to the table

## Environment Objects and States

The environment contains these objects: ["blue_block", "box", "drawer",
    ↪ "light"]

### Possible Environment Status Sets for Each Object
- drawer: ["The drawer is closed", "The drawer is open"]
- box: ["The box is closed", "The box is open"]
- light: ["The light is closed", "The light is open"]
- blue_block: ["The blue block is in the drawer", "The blue block is on
    ↪ the table", "The blue block is in the box"]

## Output Format

Output should be in this JSON format:

```json
{
    "interact_objects": ["", ...],
    "stages": [
        {
            "name": "",
                        "interacted_object": ""
            "environment": {
                // list all the interacted object environment statuses
            },
            "motions": [
                "...", // should be the motion of the interact_objects
                "..."
            ],
        },
        {
            "name": "",
```

```
                              "interacted_object": ""
               "environment": {
                   // list all the interacted object environment statuses
               },
                              "environment": {
                   // list all the interacted object environment statuses
               },
               "motions": [
                   "...",
                   "..."
               ],
           },
           ...
       ]
}
```

## Guidelines

For every task, you should split the task into stages, motions, and also
    ↪  provide the environment status during stages if there are two or
    ↪  more stages.

### Notices
- This task involves a robot arm that is initially far from all the
    ↪  objects.
    - Therefore, unless mentioned in the task goal, the first phase should
        ↪  be "reach xxx."
- The definition of a stage is: Complete interaction with an object.
    - If the task has interacted with multiple objects, the stage will
        ↪  only change during the transition between interacted objects
- Only perform behaviors that are mentioned in the task.
- In the environment for each stage, list the initial environment status
    ↪  for each object in that stage.
- The listed of "environment" object for each stage should be all of the
    ↪  objects
- "interact_objects" should only contain the object that will be
    ↪  interacted
- The adjacent interact_object should be different
- The motion in each stage can only be subset of the interacted_object's
    ↪  motion list

Example:
Task: open the drawer then open the box then move the blue block to the
    ↪  table
Initial environment:
- drawer: "The drawer is closed"
- box: "The box is closed"
- light: "The light is closed"
- blue_block: "The blue block is in the drawer"
Output:
```json
{
        "interact_objects": ["drawer", "box", "blue_block"],
        "stages": [
                {
                        "name": "open the drawer",
                        "interated_object": "drawer"
                        "environment": {
                                "drawer": "The drawer is closed",
                                "box": "The box is closed",
                                "light": "The light is closed",
                                "blue_block": "The blue block is in the
                                    ↪  drawer"
                        }
```

```
                        "motions": [
                                "reach the closed drawer holder top",
                                "pull the drawer out"
                        ],
                },
                {
                        "name": "open the box",
                        "interated_object": "box"
                        "environment": {
                                "drawer": "The drawer is open",
                                "box": "The box is closed",
                                "light": "The light is closed",
                                "blue_block": "The blue block is in the
                                    ↪ drawer"
                        }
                        "motions": [
                                "reach the box holder back",
                                "slide the box holder forward"
                        ],
                },
                {
                        "name": "move the blue block to the table",
                        "interated_object": "blue_block"
                        "environment": {
                                "drawer": "The drawer is open",
                                "box": "The box is open",
                                "light": "The light is closed",
                                "blue_block": "The blue block is in the
                                    ↪ drawer"
                        },
                        "motions": [
                                "reach the blue block",
                                "lift the grasped blue block",
                                "move the blue block to the top of the
                                    ↪ table"
                                "place the blue block down to the table"
                        ]
                }
        ]
}
```

Task: move the blue block to the table
Initial environment:
- drawer: "The drawer is open"
- box: "The box is open"
- light: "The light is open"
- blue_block: "The blue block is in the drawer"
Output:
```json
{
        "interact_objects": ["blue_block"],
        "stages": [
                {
                        "name": "move the blue block to the table",
                        "interated_object": "blue_block"
                        "environment": {
                                "drawer": "The drawer is open",
                                "box": "The box is open",
                                "light": "The light is open",
                                "blue_block": "The blue block is in the
                                    ↪ drawer"
                        },
                        "motions": [
                                "reach the blue block",
```

```
                                    "lift the grasped blue block",
                                    "move the blue block to the top of the
                                       ↪ table"
                                    "place the blue block down to the table"
                        ]
                }
        ]
}
```

=====

Task: {{task_name}}
Initial environment:
- drawer: {{initial_drawer}}
- box: {{initial_box}}
- light: {{initial_light}}
- blue_block: {{initial_block}}
Output:
```

Listing 1: Prompt for Task Knowledge Generation

## BROADER IMPACT

Our research is focused on developing a reward model for manipulation tasks, aiming to accelerate the adoption of robotic applications in complex settings. While we haven't identified any immediate negative impacts or ethical concerns, it's crucial that we remain vigilant and continuously assess any potential societal implications as our work evolves.

