# OpenReview forum: "VICtoR: Learning Hierarchical Vision-Instruction Correlation Rewards for Long-horizon Manipulation"
_ICLR.cc/2025/Conference — ICLR 2025 Poster_

### Official Review · Reviewer_Hvoi · 2024-10-30

**Soundness:** 3
**Presentation:** 3
**Contribution:** 3
**Rating:** 6
**Confidence:** 3

**Summary:**

This paper proposes VICtoR, a hierarchical reward model for long-horizon manipulation tasks that learns from action-free videos and language instructions. The proposed approach addresses challenges in task complexity, sub-stage awareness, and object state estimation through a stage detector and motion progress evaluator. VICtoR outperforms existing models by using a novel stage detector and motion progress evaluator, achieving a 43% improvement in success rates for complex tasks.

**Strengths:**

- The paper effectively advances research by integrating LLMs for high-level hierarchical planning, which allows the model to assess the robot’s progress at each step.
- It incorporates an innovative approach to extract relevant information from short video clips, enabling the system to understand key aspects of the task without viewing the entire sequence.
- The methodology section is well-structured, clearly outlining the three core components of the proposed approach, aiding reader comprehension.
- Results in the experiments section demonstrate substantial improvements over previous methods in complex task settings, showcasing the effectiveness of the hierarchical framework.
- Reward visualizations effectively illustrate the approach’s ability to accurately assess performance.

**Weaknesses:**

- A more thorough evaluation of each individual component would clarify its contribution to the overall system’s performance.
- Additional evaluations on zero-shot capabilities, specifically with unseen trajectories, would be valuable in understanding the approach’s generalization.
- The paper lacks an assessment of the computational cost and resource requirements of the proposed method.
- The reliance on GPT-4 could pose potential challenges, which are not thoroughly addressed; analyzing GPT-4 outputs in greater detail could reveal possible limitations.
- A comparative discussion of other methods for long-horizon manipulation, such as hierarchical reinforcement learning (HRL) and task and motion planning (TAMP), is missing and would have enriched the analysis.
- While the authors mention using data from XSkill3 for real-world experiments, they do not provide sufficient details on the dataset’s size, diversity, or the types of tasks involved.

**Questions:**

Refer to the Weaknesses section.

---

> ### Author Response · Authors · 2024-11-17
>
> Thank you for your insightful comments and your recognition of our work! Our responses below are dedicated to addressing your questions and concerns.
>
> **[W1] A more thorough evaluation of each individual component would clarify its contribution to the overall system’s performance.**
>
> Thank you for your valuable suggestion. To better evaluate the contribution of each component in VICtoR, we conducted an ablation study on the different hierarchical rewards, as presented in **Table 3**. Additionally, we assessed the robustness and accuracy of GPT-4, with detailed results provided in **Appendix B.2**.
>
> As further suggested by Reviewer D5gq, we performed an additional experiment on the three contrastive objectives for the *open box then pick block* task. The results are summarized in the table below and has been included in Appendix B.3 of the updated paper.
>
> | Loss Applied | Success rate ($\uparrow$) |
> | --- | --- |
> | All three losses | 96% |
> |$\mathcal{L}_{lfcn}$ +$\mathcal{L}\_{mcn}$ | 94% |
> |$\mathcal{L}_{lfcn}$ +$\mathcal{L}\_{tcn}$ | 64% |
> |$\mathcal{L}_{mcn}$ +$\mathcal{L}\_{tcn}$ | 0% |
>
>
> **[W2] Additional evaluations on zero-shot capabilities, specifically with unseen trajectories, would be valuable in understanding the approach’s generalization.**
>
> Thank you for your valuable suggestion. We would like to emphasize that VICtoR provides a more generalized approach to reward generation compared to previous VIC methods. To further demonstrate VICtoR’s strengths, we present a comparison of the required training resources and generalization capabilities of VICtoR and other VIC methods in the table below.
>
> As shown in the table, VICtoR can be applied to tasks involving any combination of motions in the target environment, whereas other VIC methods are limited in this regard. The table highlights VICtoR’s superior generalization capabilities across varying task complexities and lengths.
>
> |  | Pre-trained Envs | Target Env | Reward Generalization Ability  |
> | -------- | -------- | -------- | -------- |
> | VICtoR | X     | Motion-level demo     | Same env, any task composed of seen motion |
> | LIV etc. | Human demo.     | **FULL-length demo**| X     |
>
>
> **[W3] The paper lacks an assessment of the computational cost and resource requirements of the proposed method.**
>
> Thank you for pointing this out. Below, we summarize the details of our method's computational cost and resource requirements:
> Training the motion progress evaluator requires approximately 2 GPU hours. For RL policy training, querying our reward model for 20,000 steps takes around 12 GPU hours. Both were tested on a single RTX 4090.
> We also measured the inference time of VICtoR. The object detector takes approximately 51.5 ms, while the reward model accounts for 24 ms on average over 1,000 steps (tested on an Intel(R) Core(TM) i7-13700 CPU and an RTX 3090 GPU). Given that end-effector action latency is typically significant in both simulation and real-world scenarios, we believe the 0.08-second inference time would not adversely affect the agent's efficiency, even in real-world applications.
> We have included this information in the updated paper.

---

> ### Author Response · Authors · 2024-11-17
>
> **[W4] The reliance on GPT-4 could pose potential challenges, which are not thoroughly addressed; analyzing GPT-4 outputs in greater detail could reveal possible limitations.**
>
> Building on the findings in [1], our approach leverages the strengths of LLMs in commonsense reasoning, effectively utilizing them for task decomposition while minimizing the risk of inaccuracies. In Appendix B.2, we demonstrate that large language models (LLMs) exhibit high accuracy in handling these types of tasks. Specifically, we evaluate GPT-4's ability to decompose tasks into sequences of motions, achieving a 96% success rate. These positive results highlight how the task knowledge derived from GPT-4 ensures the robustness and precision for our reward models. If there are any concerns, we welcome further discussion and are prepared to conduct additional experiments to address them comprehensively.
>
> [1] Wenlong et al., Language Models as Zero-Shot Planners: Extracting Actionable Knowledge for Embodied Agents, ICML 2022
>
>
> **[W5] A comparative discussion of other methods for long-horizon manipulation, such as hierarchical reinforcement learning (HRL) and task and motion planning (TAMP), is missing and would have enriched the analysis.**
>
> We **have compared** our work with HRL and TAMP, as summarized in **Related Work** and **Appendix E**. If there are specific works that the reviewer believes should be included to make the comparison more comprehensive, please kindly provide the references. We will review them and incorporate them into our paper as appropriate.
>
> **[W6] While the authors mention using data from XSkill for real-world experiments, they do not provide sufficient details on the dataset’s size, diversity, or the types of tasks involved.**
>
> Thank you for pointing this out. We have included key information about the XSKILL dataset in **Figure 3**, such as the number of interactable objects, the number of tasks, the total number of videos, and the type of demonstrations. Nevertheless, we are happy to provide more detailed descriptions of each task in the updated paper.
>
>
> Thank you again for your time and dedication in reviewing our paper. If you have any unresolved concerns after reading our responses, please let us know. We look forward to learning more from you during our constructive discussions.

---

> ### Author Response · Authors · 2024-11-22
>
> Dear Reviewer Hvoi,
>
> Thank you once again for your time and dedication in reviewing our paper.
>
> Based on your valuable comments, we have conducted additional experiments and revised the manuscript. Since the reviewer-author discussion ends in **five** days, we would like to know if our responses and revisions have addressed your concerns. If so, we kindly request that you reflect this in your evaluation scores. If any points remain unclear, we are eager to continue the discussion and provide further details or explanations.
>
> We sincerely hope that, through our joint efforts, we can make this work more robust and impactful.
>
> Thank you very much.

---

> > ### Comment · Reviewer_Hvoi · 2024-11-27
> >
> > The authors response clarify my concerns.

---

> > > ### Author Response · Authors · 2024-11-27
> > >
> > > Thank you for confirming that the concerns have been resolved.
> > >
> > > We would greatly appreciate it if the reviewer could reflect this in their evaluation score or express their support for our work during the upcoming reviewer-AC discussion. Additionally, if there is any further evidence the reviewer would like to see in order to reconsider their score, please feel free to let us know.
> > >
> > > We will carefully revise the paper following your guidance during our conversations.
> > >
> > > Thank you again for reviewing our paper.

---

### Official Review · Reviewer_R2Xa · 2024-11-02

**Soundness:** 3
**Presentation:** 3
**Contribution:** 2
**Rating:** 6
**Confidence:** 4

**Summary:**

This paper introduces a hierarchical vision-instruction correlation method to estimate rewards for long-horizon robotic manipulation tasks. Specifically, the approach initially identifies task-aware objects and employs GPT-4V to associate the object status with the corresponding motion stage. Additionally, this paper proposes a contrastive learning-based method that evaluates the reward of visual observations by calculating the similarity between visual observations and language instructions. Experimental results validate the efficacy of this method.

**Strengths:**

1. This paper proposes a hierarchical reward estimation method to provide vision-based reward for long-horizon manipulation tasks.
2. This paper describes this method clearly.
3. The simulation and real-world experiments prove the effectiveness of this method.

**Weaknesses:**

1. The method requires lots of relevant data for estimating visual object information, which makes it struggle to generalize to new task scenarios due to reliance on the object status classifier. Could this method generalize to any novel task?

2. The comparison results are unfair. This method utilizes the robotic manipulation videos of the same tasks to train the contrastive learning method, while the compared methods such as LIV are pretrained encoders.  Would these vision-language correlation methods achieve comparable results when the encoder is fine-tuned on the target tasks?

3. The conservative learning method for vision-language reward estimation is common. The stage and motion detection method seems to simply achieve state-based rewards through visual models by overfitting on specific manipulation tasks.

**Questions:**

Please refer to the weakness.

My questions are:
1. Could this method generalize to novel tasks?
2. Could other methods achieve comparable results when fine-tuned on the target task?

---

> ### Author Response · Authors · 2024-11-17
>
> Thank you for your insightful and detailed comments! Our responses below are dedicated to addressing your questions and concerns.
>
> **[W1] The method requires lots of relevant data for estimating visual object information, which makes it struggle to generalize to new task scenarios due to reliance on the object status classifier.**
>
> Thank you for raising this concern. We can actually decouple the object state detection process into **object detection** and **object status estimation**.
>
> Regarding object detection, we believe the generalization ability of off-the-shelf object detectors is not a concern. For example, in our experiments on the XSkill dataset, we applied MDETR **without any domain-specific fine-tuning** and still achieved accurate bounding box detection.
>
> Regarding object status estimation, we acknowledge that data collection for training is indeed required. However, we would like to emphasize that such data can now be easily collected. For example, modern LVLMs (e.g., GPT-4V) can be used to automatically annotate the object status of cropped images produced by the object detector mentioned above. We have conducted an ablation study on automatic annotation, and the results are summarized in **Appendix B.4** of the updated paper.
>
> Thank you again for raising this concern.
>
> **[W1] Could this method generalize to any novel task? [Q1] Could this method generalize to novel tasks?**
>
> Compared to prior VIC methods, VICtoR offers a more generalized approach to reward generation. To highlight VICtoR’s strengths, we compare the required training resources and generalization capabilities of VICtoR and other VIC methods in the table below. As shown in the table, our VICtoR can be applied to tasks involving any combination of motions in the target environment, whereas other VIC methods cannot.
>
> |  | Pre-trained Envs | Target Env | Reward Generalization Ability  |
> | -------- | -------- | -------- | -------- |
> | VICtoR | X     | Motion-level demo     | Same env, any task composed of seen motion |
> | LIV etc. | Human demo.     | **FULL-length demo**| X     |
>
> **[W2] The comparison results are unfair. This method utilizes the robotic manipulation videos of the same tasks to train the contrastive learning method, while the compared methods such as LIV are pretrained encoders. Would these vision-language correlation methods achieve comparable results when the encoder is fine-tuned on the target tasks? [Q2] Could other methods achieve comparable results when fine-tuned on the target task?**
>
> As stated in lines 804–809, **all methods** are trained or fine-tuned in the target environment to ensure a fair comparison. As shown in the table in W1, existing VIC methods require extensive demonstrations in other environments for pre-training and **full-length** demonstrations in the target environment for fine-tuning. In contrast, our VICtoR requires only motion-level demonstrations in the target environment for training.
>
>
> **[W3] The conservative learning method for vision-language reward estimation is common. The stage and motion detection method seems to simply achieve state-based rewards through visual models by overfitting on specific manipulation tasks.**
>
> We acknowledge that InfoNCE is a well-known contrastive objective; however, we argue that our insights into applying InfoNCE to effectively assess task progress for long-horizon tasks are valuable, rather than merely constructing an arbitrary combination of three losses.
>
> The aim of VIC reward models is to address the lack of well-defined reward functions in real-world environments. Without these, RL methods may become infeasible, especially for long-horizon tasks. In this work, we highlight three key challenges that prior VIC methods often encounter. To address these challenges, we introduce novel **contrastive loss applications** (temporal, motion, and progress) in our motion progress evaluator to better capture the nuances of task progress. Consequently, our approach generates more precise rewards, leading to superior evaluation outcomes, as demonstrated in Tables 1 and 2.
>
> Thank you again for your time and dedication in reviewing our paper. If you have any unresolved concerns after reading our responses, please let us know. We look forward to learning more from you during our constructive discussions.

---

> ### Author Response · Authors · 2024-11-22
>
> Dear Reviewer R2Xa,
>
> Thank you once again for your time and dedication in reviewing our paper.
>
> Based on your valuable comments, we have conducted additional experiments and revised the manuscript. Since the reviewer-author discussion ends in **five days**, we would like to know if our responses and revisions have addressed your concerns. If so, we kindly request that you reflect this in your evaluation scores. If any points remain unclear, we are eager to continue the discussion and provide further details or explanations.
>
> We sincerely hope that, through our joint efforts, we can make this work more robust and impactful.
>
> Thank you very much.

---

> > ### Comment · Reviewer_R2Xa · 2024-11-27
> > **Thanks for this response**
> >
> > This response addressed my concerns, and I will raise the score to 6.

---

> > > ### Author Response · Authors · 2024-11-27
> > >
> > > Thank you for informing us that the concerns have been addressed and for raising the score!
> > >
> > > We would greatly appreciate it if you could express your support for our work during the upcoming reviewer-AC discussion. Additionally, if there is any further evidence you would like us to provide to reconsider the score, please feel free to let us know.
> > >
> > > We will carefully revise the paper based on your guidance from our conversations.
> > >
> > > Thank you again for reviewing our paper.

---

### Official Review · Reviewer_sb4X · 2024-11-03

**Soundness:** 3
**Presentation:** 3
**Contribution:** 2
**Rating:** 6
**Confidence:** 4

**Summary:**

This paper proposes a method called VICtoR that utilizes LLM (GPT-4) and VLM (CLIP) to calculate the task rewards for long-horizon robotic manipulation. Specifically, it decomposes the language instructions to different stages with the help of GPT-4, and uses MDETR to detect objects and classify their states. Comparing the states with the desired states from GPT-4, the algorithm can determine if the stage has been completed. For the motions inside each stage, VICtoR trains the CLIP encoders with multiple contrastive losses to learn the task progress corresponding to the instruction. Combined with the above reward, a model is trained with self-constructed benchmarks and shows better performance compared to multiple baselines.

**Strengths:**

1. The method is intuitive. Decomposing a long-horizon task with LLM and utilizing VLM for the short-term progress evaluator are both reasonable and have been validated in previous works.
1. The visualizations are intuitive. I can easily get the idea of most figures and they help to understand the paper. The reward curves and t-SNE graph prove the effectiveness of the proposed reward design and contrastive loss designs.
1. The results shown in the paper are good, compared to other baseline methods.

**Weaknesses:**

1. The technical novelty is relatively limited. The proposed method mainly has two parts, LLM for decomposing the long-horizon task and CLIP for the short-term progress evaluator. These two parts have both been extensively explored. There have been multiple related works mentioned in the paper. Even in Line 186, the authors acknowledged that the differences are relatively nuanced.
1. The experiments are conducted on a self-collected dataset, not the widely adopted CALVIN benchmark. As mentioned in Line 343-346, the proposed method has critical limitations when dealing with some meticulous actions like rotations.
1. Some technical details are confusing and need further clarification. Please see the questions part below.
1. Minors. Typo in Line 143 (as as)

**Questions:**

1. Line 160, how are the objects states autonomously annotated during real-world experiments?
1. Line 205-208, What if the object detector fails? Will using cropped images for object states classification have certain problems? For example, some object states are positional relationships that need a larger visual field to accurately determine them. What is the model structure for the classifier P?
1. Line 471, why the curves are called ``potential curves''? Are the rewards calculated with collected videos? Do you use the rewards for action rollouts?
1. How do you deal with the long inference time of LLM, when you are training the policy with the proposed design?

---

> ### Author Response · Authors · 2024-11-17
>
> Thank you for your insightful and detailed comments! Our responses below are dedicated to addressing your questions and concerns.
>
> **[W1] The technical novelty is relatively limited. The proposed method mainly has two parts, LLM for decomposing the long-horizon task and CLIP for the short-term progress evaluator. These two parts have both been extensively explored. There have been multiple related works mentioned in the paper. Even in Line 186, the authors acknowledged that the differences are relatively nuanced.**
>
> Thank you for raising this concern. We would like to clarify that our work offers significant technical contributions. While we acknowledge the extensive application of LLMs and VLMs in various research areas, VIC reward generation is an emerging topic, and we are the **first** to design a VIC method specifically for long-horizon tasks.
>
> Our work focuses on establishing what constitutes an effective VIC reward model for such tasks. We conducted an extensive review of prior methods, identifying three major challenges, as outlined in Figure 1, which they often fail to address effectively. To overcome these, we proposed a novel **hierarchical reward model** tailored to these challenges, achieving substantial improvements in our experiments. Our use of LLMs, VLMs, and contrastive objectives is informed by solid pilot studies and unique insights, rather than being an arbitrary combination of existing tools.
>
> Regarding the statement in Line 186, the term “nuanced” highlights our approach to leveraging LLMs for generating more fine-grained task information. This includes not only decomposing tasks into stages but also specifying desired object statuses and required motions, demonstrating the LLM's deeper understanding of the environment.
>
> **[W2] The experiments are conducted on a self-collected dataset, not the widely adopted CALVIN benchmark. As mentioned in Line 343-346, the proposed method has critical limitations when dealing with some meticulous actions like rotations.**
>
> Thank you for raising this concern. The issue mentioned in Lines 343-346 applies to **all VIC methods**, as they rely solely on visual inputs and instructions to generate rewards, without leveraging oracle-provided environmental information. One potential solution is to use wrist-camera image inputs; however, this approach has not been utilized in prior methods and could lead to issues of unfair comparison.
> Additionally, the CALVIN benchmark is **highly challenging for RL methods**, as evidenced by its leaderboard. Most approaches on this benchmark rely on imitation learning with sophisticated components, such as diffusion models. Comparing methods on this benchmark might **obscure the distinctions** between VIC approaches.
> Nevertheless, we acknowledge the importance of enabling VIC methods to provide reward signals for a variety of tasks. We are currently investigating comparisons on other well-established benchmarks.
>
> **[W4] Minors. Typo in Line 143 (as as)**
>
> Thank you for your careful review. We have corrected the issue and thoroughly inspect the paper to ensure there are no remaining typos.

---

> ### Author Response · Authors · 2024-11-17
>
> **[Q1] Line 160, how are the objects states autonomously annotated during real-world experiments?**
>
> For XSkill, the object status label can be automatically labeled as the data in XSkill are all consecutive collected and are labeled with unit-skill, which enables us to mark the change that occurs in a single skill and inherit the status of the non-interacted object from previous actions. To make better adaptations to other datasets, as object detection is a well-studied problem in the computer vision field, we can easily employ the state-of-the-art LVLM or VQA model, such as GPT-4V, to help on the automated annotation of object states.
>
> **[Q2] Line 205-208, What if the object detector fails? Will using cropped images for object states classification have certain problems? For example, some object states are positional relationships that need a larger visual field to accurately determine them. What is the model structure for the classifier P?**
>
> - As stated in Lines 216-217, if the set of estimated object statuses does not match any stage's object statuses, it is reset to the initial stage to avoid misleading the policy's optimistic judgment about task progress.
> - The question about the visual field is indeed critical. In our evaluation, we include tasks such as placing a blue block into a drawer, which requires a visual field larger than the block's original bounding box. To address this, we enlarge the detected bounding boxes by a fixed width to better capture the relationship between the object and its surrounding area.
> - The structure of classifier P includes a CLIP Text Encoder for processing object state descriptions and a CLIP Image Encoder for processing cropped images. These encoders are followed by a downstream MLP classifier that scores relevance. This information is missing in the paper; we have included  it  in the new version for clarity. Thank you for pointing this out.
>
>
> **[Q3] Line 471, why the curves are called ``potential curves''? Are the rewards calculated with collected videos? Do you use the rewards for action rollouts?**
>
> - Recall that our VICtoR is a VIC method. Thus, during the experiment shown in Figure 5, all methods generate rewards by taking a task instruction and either corresponding or irrelevant videos as inputs. A desired behavior is for the reward model to generate negative potential to indicate a mismatch between the instruction and the video (only our VICtoR achieves this). Additionally, our model is trained merely on video demonstrations **without any action vectors and ground truth environmental observation vectors**. In the inference process, our model also generates rewards from task descriptions and visual observations **without any action information** included.
> - Regarding the term "potential," it refers to a classical type of reward function in the RL literature, first introduced in 1999. In potential-based rewards, "potential" refers to a scalar function that represents the relative desirability or progress of states in the environment. The calculation of rewards from potential is detailed in Eq. 9 (Line 329). More information can also be found in our cited reference [45].
>
>
> **[Q4] How do you deal with the long inference time of LLM, when you are training the policy with the proposed design?**
>
>
> Thank you for the important question. In our pipeline, once the task is determined (i.e., the task instruction is received), the LLM only needs to decompose the long-horizon task once before policy training begins. Therefore, our pipeline does not suffer from the long inference time of the LLM.
>
> Thank you again for your time and dedication in reviewing our paper. If you have any unresolved concerns after reading our responses, please let us know. We look forward to learning more from you during our constructive discussions.

---

> ### Comment · Reviewer_sb4X · 2024-11-18
> **Response to the Rebuttal**
>
> Thanks for the reply. My questions [Q1-Q4] have been clarified. I have also read other reviewers' comments and authors' feedback. I think we share similar concerns about this paper. The benchmark is relatively limited and it would be better to see the method's generalization ability.
> - The paper mainly utilizes the XSkill dataset, which is relatively simple compared to benchmarks like CALVIN. Personally speaking, the authors don't need to worry about unfair comparisons with BC-based methods, as long as the method makes certain contributions based on fairly compared baselines.
> - The method needs labels for objects for training. XSkill provides labels for objects. Though the authors claim that some CV methods can be applied to achieve autonomous annotation, I think more experimental results are better to illustrate that.
> - When people use LLMs and VLMs, they hope these foundation models can be well adapted to unseen environments and have zero-shot ability. I think the high-level comparisons of VICtoR to LIV etc need further quantitative demonstrations.

---

> ### Author Response · Authors · 2024-11-20
> **Response to the Reviewer's follow-up comments**
>
> Thank you for your prompt reply. Please find our responses:
>
> **1. XSkill vs. CALVIN Benchmark Comparison:**
>
> We are unsure why XSkill is considered "relatively simple" compared to CALVIN. Below, we compare the two benchmarks:
>
> | Dataset | Venue   | # of interactive objects | Maximum stage of tasks | Scene type       | Scene attribute (background/ Light effect/ shadow) | object interaction (e.g., block in slider or cloth in sink) | rotation movement |
> |---------|---------|--------------------------|------------------------|------------------|----------------------------------------------------|---------------------------------------------------|-------------------|
> | CALVIN  | RA-L’22 | 6                        | 5                      | Simulated indoor | X/single light source/hard shadow                                              | V                                                 | V                 |
> | XSkill  | CoRL’23 | 4                        | 4                      | Real Kitchen     | V/real/soft shadow                                              | V                                                 | V                 |
>
> Although the CALVIN benchmark contains more interactable objects and slightly longer tasks, its scene attributes are oversimplified. Considering that our research focuses on VIC reward generation, we believe our real-world experiments on the XSkill dataset provide sufficient significance. Nonetheless, we will continue exploring opportunities to evaluate VIC methods on other benchmarks.
>
> We also attempted to train RL policies using different VIC reward methods on CALVIN, but these consistently performed poorly. We attribute this to the camera settings in CALVIN, which make it challenging for VIC methods to observe the effects of certain actions (e.g., rotation). While incorporating visual inputs from additional viewpoints could be a potential solution, it may raise concerns of unfair comparison, as prior VIC methods do not leverage such additional inputs.
>
> Regarding the comparison with BC methods, we respectfully maintain that such a comparison is not appropriate for our work. BC methods require actions to train the policy, which is fundamentally different in nature from our approach. Our research focuses on developing the VIC reward model. In the VIC literature, the established practice is to compare the performance of the same RL method trained with different VIC reward models. This is exactly what we have done in our paper.
>
> **2. Experiment with LVLM for Object Status Annotation:**
>
> Thank you for your clear suggestion. We have conducted an ablation study and included the results in the updated paper. Further details can be found in the general response.
>
> **3. Generalization Ability Comparison:**
>
> As presented in our response to Reviewer Hvoi's W2, our VICtoR method can be applied to **tasks involving any motion combination** within the same environment, whereas other VIC methods **cannot**. We consider this a significant step towards greater generalization ability for reward models. In practice, a common scenario involves requiring a robot to perform different tasks in the same environment (e.g., a household setting).
>
> Additionally, to the best of our knowledge, there is **no existing VIC method with zero-shot capability** to generate rewards in unseen environments without requiring any environmental information. If the reviewer can provide a specific reference, we would be happy to review it and include a comparison.

---

> > ### Comment · Reviewer_sb4X · 2024-11-22
> > **Response to the Authors**
> >
> > Thanks for the further clarifications. Sorry for the confusion about my expression of "relatively simple". I would like to mean that CALVIN has more well-performing baselines and several challenging tasks like rotations. In terms of zero-shot capability, it would be great to explore this subject, considering the original generalization ability of LLM and VLMs adopted in VIC methods. Anyway, thanks for the rebuttal and I am pleased to increase my score.

---

> > > ### Author Response · Authors · 2024-11-22
> > >
> > > Thank you for your additional comments and detailed explanations.
> > >
> > > - Regarding the benchmark, we fully agree that conducting experiments on well-established benchmarks makes the evaluation more convincing. That is why we evaluate VIC methods on both self-constructed simulated environments and a well-known real-world benchmark, XSkill. We will continue conducting experiments on various benchmarks and tasks, provided they align with our research goals.
> > >
> > > - Regarding generalization ability, advancing VIC methods' generalization, including zero-shot performance in unseen environments, is indeed a valuable direction. As shown in the table, our VICtoR takes a significant step toward improved generalization. Nevertheless, we will certainly consider the direction you suggested and plan to conduct such studies in the future.
> > >
> > > We sincerely thank you again for your timely and constructive discussions, which have made our work more robust and impactful. We also appreciate your willingness to increase the score. We will continue exploring opportunities to compare VIC methods on more benchmarks. If you have any further suggestions, please feel free to share them with us.

---

### Official Review · Reviewer_D5gq · 2024-11-04

**Soundness:** 2
**Presentation:** 3
**Contribution:** 3
**Rating:** 6
**Confidence:** 4

**Summary:**

The authors propose a hierarchical reward model that provides reward signals for long-horizon manipulation tasks. First, It proposes using a Task Knowledge Generator to decompose a task into stages and identify the necessary object states and motions for each stage. It then adopts a Stage Detector to detect the stage and object states given the visual observations. Finally, with a Motion Progress Evaluator that assesses motion completion within stages, the framework gives the overall reward for the long-horizon task. While the task knowledge generator leverages LLMs, the other parts are trained solely on primitive motion demonstrations. The paper shows that their proposed framework produces better rewards for long-horizon manipulation tasks, thus facilitating policy learning with methods such as PPO.

**Strengths:**

+ The approach explores how to leverage LLMs for providing task decomposition for long-horizon manipulation tasks by defining stages and motions as the task hierarchy. And it seems that LLMs such as GPT-4 work well for this setup.
+ The design of the modules sounds reasonable, i.e., leverage MDETR for stage and object status detection and leverage CLIP for motion progress evaluator. The training losses are well-motivated. The results are good.

**Weaknesses:**

- The long-horizon tasks used to evaluate the proposed framework are a bit limited, e.g., even compared to a popular benchmark [1] proposed in 2020. A more diverse set of tasks would help to fully verify the usefulness and adaptability of this approach.

[1] ALFRED: A Benchmark for Interpreting Grounded Instructions for Everyday Tasks

**Questions:**

+ I didn't find ablation studies of the three contrastive losses used in training the motion progress evaluator. It would be better to know their contributions to the overall quality of the generated reward.
+ The proposed approach utilizes motion decomposition of subtasks, a concept similar to the ones used in [2]. It would be helpful to add a discussion about it. In particular, I wonder will the proposed approach also works well at a potentially more granular motion level (e.g., a subskill in the Move Chair task in [2]).

[2] Chain-of-Thought Predictive Control

---

> ### Author Response · Authors · 2024-11-17
>
> Thank you for your insightful comments and your recognition of our work! Our responses below are dedicated to addressing your questions and concerns.
>
> **[W1] The long-horizon tasks used to evaluate the proposed framework are a bit limited, e.g., even compared to a popular benchmark [1] proposed in 2020. A more diverse set of tasks would help to fully verify the usefulness and adaptability of this approach.**
>
> Thank you for your valuable suggestion. However, we found that the action space in ALFRED is too coarse for evaluating VIC methods in manipulation tasks. For example, actions like *PickupObject* in ALFRED immediately attach the object to the robot's gripper, lacking the necessary granularity for fine-grained evaluation.
>
> In our work, we tested all VIC methods in both simulated and real-world experiments. Our simulated environments were meticulously designed to include key components for comprehensive evaluation, such as long-horizon tasks involving chain-of-action subtasks where the success of one step directly impacts subsequent steps. Many existing long-horizon benchmarks lack this critical aspect. Additionally, we evaluated VICtoR on the real-world XSkill dataset to better demonstrate its practical applicability.
>
> While other benchmarks, such as CALVIN, exhibit some properties of interest, our experiments revealed that certain actions, such as rotations, are difficult to discern from its visual inputs. This limitation makes it challenging to effectively differentiate the performance of VIC methods on CALVIN. Nonetheless, we are actively exploring ways to compare VIC methods on other existing benchmarks.
>
> **[Q1] I didn't find ablation studies of the three contrastive losses used in training the motion progress evaluator. It would be better to know their contributions to the overall quality of the generated reward.**
>
> Thank you for your precious suggestion. We have conducted an ablation study on the three contrastive loss functions, as shown in the table below. The results demonstrate that each of the three loss designs contributes to the final outcome, each serving a distinct function. We have included this new experiment in Appendix B.3.
>
> | Loss Applied | Success rate ($\uparrow$) |
> | --- | --- |
> | All three losses | 96% |
> |$\mathcal{L}_{lfcn}$ +$\mathcal{L}\_{mcn}$ | 94% |
> |$\mathcal{L}_{lfcn}$ +$\mathcal{L}\_{tcn}$ | 64% |
> |$\mathcal{L}_{mcn}$ +$\mathcal{L}\_{tcn}$ | 0% |
>
> **[Q2] The proposed approach utilizes motion decomposition of subtasks, a concept similar to the ones used in [2]. It would be helpful to add a discussion about it. In particular, I wonder will the proposed approach also works well at a potentially more granular motion level (e.g., a subskill in the Move Chair task in [2]).**
>
> Thank you for your valuable suggestion. We havel updated our paper by including a comparison between the prior method on motion decomposition and ours in Appendix E.
>
> In the CoTPC work, their subskills are analogous to our defined motions, breaking tasks into components such as reaching and adjusting. Their focus is on training a model to predict the next subskill, employing a “chain-of-thought” control strategy to enhance accuracy. In contrast, our approach decomposes tasks into stages, motions, and progress, providing a more comprehensive evaluation of the robot's overall task progress and enabling more accurate reward generation.
>
> Furthermore, our setting is fundamentally different from theirs: we work exclusively with **action-free videos** paired with language instructions. This distinguishes our method from CoTPC, which leverages actions to group subskills.
>
>
> Thank you again for your time and dedication in reviewing our paper. If you have any unresolved concerns after reading our responses, please let us know. We look forward to learning more from you during our constructive discussions.

---

> ### Author Response · Authors · 2024-11-22
>
> Dear Reviewer D5gq,
>
> Thank you once again for your time and dedication in reviewing our paper.
>
> Based on your valuable comments, we have conducted additional experiments and revised the manuscript. Since the reviewer-author discussion ends in **five days**, we would like to know if our responses and revisions have addressed your concerns. If so, we kindly request that you reflect this in your evaluation scores. If any points remain unclear, we are eager to continue the discussion and provide further details or explanations.
>
> We sincerely hope that, through our joint efforts, we can make this work more robust and impactful.
>
> Thank you very much.

---

> > ### Comment · Reviewer_D5gq · 2024-11-26
> > **Thanks for the author response**
> >
> > The author response has addressed my concern.

---

> > > ### Author Response · Authors · 2024-11-27
> > >
> > > Thank you for confirming that the concerns have been resolved.
> > >
> > > We would greatly appreciate it if the reviewer could reflect this in their evaluation score or express their support for our work during the upcoming reviewer-AC discussion. Additionally, if there is any further evidence the reviewer would like to see in order to reconsider their score, please feel free to let us know.
> > >
> > > We will carefully revise the paper following your guidance during our conversations.
> > >
> > > Thank you again for reviewing our paper.

---

### Author Response · Authors · 2024-11-17
**General Response to All Reviewers**

We would like to express our sincere gratitude to all the reviewers for their valuable comments and suggestions. We are pleased that the strengths of our work have been recognized by the reviewers:

- The hierarchical reward model design and training methods for long-horizon tasks are well-motivated, reasonable and intuitive **(D5gq, sb4X, R2Xa, Hvoi)**.
- The methodology is clearly described and well-structured, making it easy to understand and follow **(R2Xa, Hvoi)**.
- The paper demonstrates substantial empirical benefits, achieving strong results in both simulation and real-world experiments, validating the approach’s effectiveness **(R2Xa, Hvoi, sb4X)**.
- Innovative extraction of task-relevant information from short video clips enables efficient task understanding without viewing the entire sequence **(Hvoi)**.
- The visualizations, including reward curves and t-SNE graphs, effectively demonstrate the method’s efficacy and aid in understanding the proposed approach **(sb4X, Hvoi)**.

Regarding the concerns raised from each reviewer, we have carefully considered each comment and made corresponding modifications (highlighted in blue) in the updated main paper, as summarized below:
- Appendix B.3 Ablation study on three contrastive losses
- Appendix C.2 Model structure of the classifier P
- Appendix C.5 Computation cost and resource requirements
- Appendix C.7 Details of experiments on Xskill
- Appendix E Motion Decomposition

---

> ### Author Response · Authors · 2024-11-20
> **Follow-up Experiment on Automatic Annotation**
>
> Dear Reviewers,
>
> As **Reviewer sb4X** suggested (and similarly raised by **Reviewer R2Xa**), we have conducted an ablation study on automatically annotating object status using modern LVLMs (GPT-4V in our case), as detailed in **Appendix B.4** of the updated paper.
>
> In this ablation study, we compare the consistency between LVLM-annotated and human-annotated labels for object images cropped using MDETR (without fine-tuning). The results demonstrate a high level of agreement between the two. Most inconsistencies occur during the fuzzy phases when switching motions, which can be easily corrected or are even negligible.
>
> Please note that our VICtoR is **not a training-free method**, and currently, **no existing VIC method** can achieve zero-shot performance in an unseen environment. However, our new experiments indicate that it is highly feasible to apply VICtoR in new environments.
>
> We hope that these results address the reviewers' concerns on this point.

---

### Author Response · Authors · 2024-11-25
**Discussion ends soon**

Dear Reviewers,

Thank you again for your time and effort in reviewing our paper.

As the discussion period will conclude in **two days**, we kindly request your assistance in reviewing whether our responses have adequately addressed your concerns. If you find that our responses have satisfactorily resolved them, we would appreciate it if you could reflect this in your evaluation scores. If not, we are more than happy to provide additional details or clarifications.

Additionally, we would like to extend our gratitude to **Reviewer sb4X** for actively participating in the discussion early on, providing constructive comments, and expressing their willingness to adjust their score. We have gained valuable insights through our conversations.

We look forward to hearing more from you.

Best,

Authors of Submission 6356

---

### Meta-Review · Area_Chair_LewH · 2024-12-17

**Metareview:**

The paper is rated positively by all reviewers (6,6,6,6). The reviewers initially raised several concerns such as limited set of tasks compared to other benchmarks (Reviewer D5gq), no results on widely adopted benchmarks (Reviewer sb4X), unfair comparisons (Reviewer R2Xa), and lack of thorough evaluation of individual components (Reviewer Hvoi). The rebuttal addressed the concerns (details below). The AC follows the recommendation of the reviewers and recommends acceptance. However, the setup is simplistic in general. Generalization to new scenes, objects, and more complex tasks would make the paper better.

**Additional Comments On Reviewer Discussion:**

The provided rebuttal led to the increase of the scores by reviewers sb4X and R2Xa. Authors provided new results for contribution of different losses and evaluation of individual components. Also, they provided clarification responses to the rest of the concerns regarding limitation of the tasks, generalization capabilities and various other issues mentioned by the reviewers. All reviewers confirmed that the concerns were adequately addressed.

---

### Decision · Program_Chairs · 2025-01-22

Accept (Poster)